# Atomic Diffusion Models for Small Molecule Structure Elucidation from NMR Spectra

**Ziyu Xiong**
Princeton University
ziyux@princeton.edu

**Yichi Zhang**
Princeton University
zycddd@princeton.edu

**Foyez Alauddin**
Princeton University
fa1073@alumni.princeton.edu

**Chu Xin Cheng**
California Institute of Technology
ccheng2@alumni.caltech.edu

**Joon Soo An**
Princeton University
ahnjunsoo@princeton.edu

**Mohammad R. Seyedsayamdost**
Princeton University
mrseyed@princeton.edu

**Ellen D. Zhong**
Princeton University
zhonge@princeton.edu

## Abstract

Nuclear Magnetic Resonance (NMR) spectroscopy is a cornerstone technique for determining the structures of small molecules and is especially critical in the discovery of novel natural products and clinical therapeutics. Yet, interpreting NMR spectra remains a time-consuming, manual process requiring extensive domain expertise. We introduce CHEFNMR (CHemical Elucidation From NMR), an end-to-end framework that directly predicts an unknown molecule's structure solely from its 1D NMR spectra and chemical formula. We frame structure elucidation as conditional generation from an atomic diffusion model built on a non-equivariant transformer architecture. To model the complex chemical groups found in natural products, we generated a dataset of simulated 1D NMR spectra for over 111,000 natural products. CHEFNMR predicts the structures of challenging natural product compounds with an unsurpassed accuracy of over 65%. This work takes a significant step toward solving the grand challenge of automating small-molecule structure elucidation and highlights the potential of deep learning in accelerating molecular discovery.

## 1 Introduction

The molecules that sustain life come in several forms: large biopolymers such as DNA, RNA, and proteins described by our genetic code, and small molecules, which form complex metabolic pathways and influence all aspects of biology. A category of small molecules, known as secondary metabolites or **natural products**, describes those that are secreted into the environment where they serve myriad functions, such as signaling and chemical warfare. Because of these roles, natural products have delivered more than half of the FDA-approved small-molecule agents, including the majority of antibiotics and antitumor drugs in current clinical use, such as penicillin, taxol, and other blockbuster drugs such as lovastatin and semaglutide [48, 14, 68, 58].

39th Conference on Neural Information Processing Systems (NeurIPS 2025).

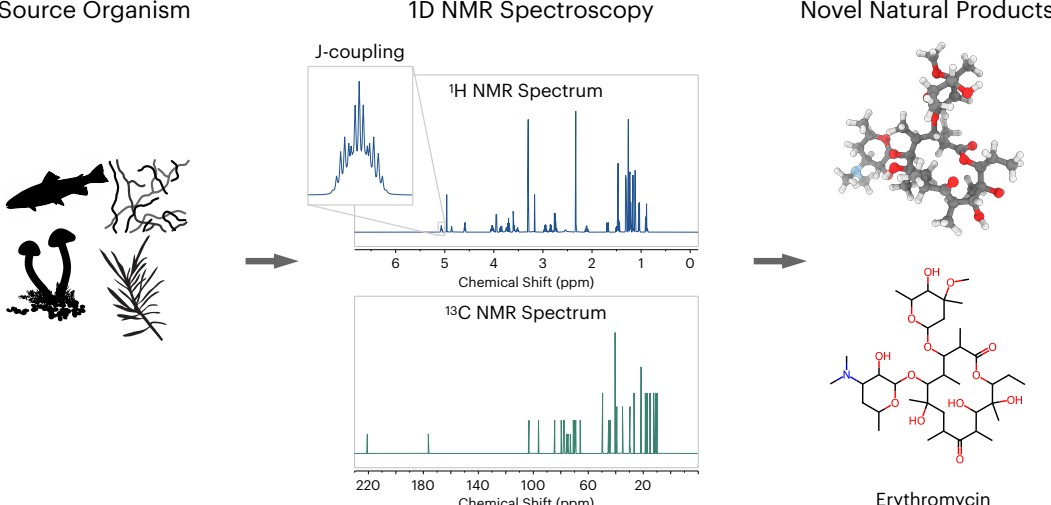

Figure 1: **Natural products** are small molecules secreted by natural sources such as plants, animals, and microorganisms (left). To identify an unknown molecule's structure, 1D NMR spectroscopy measures peaks corresponding to each proton ($^1$H) or carbon ($^{13}$C) atom (middle). The resulting *chemical shifts* (x-axis locations), *peak intensities*, and *J-coupling* (splitting patterns) encode information on chemical groups and connectivities, from which the molecular structure can be deduced (right).

The functions of small molecules are intrinsically linked to their molecular structures, which govern their chemical and biological reactivity. Very recently, deep learning methods have revolutionized the prediction of a protein's 3D structure from its amino acid sequence encoded in the genome [30, 2]. Small molecules, by contrast, are neither directly genetically encoded nor repeating polymers. Structure elucidation therefore relies on *de novo* experimental methods for every new molecule, making the discovery of cellular metabolites, essential molecules, antibiotics, and other therapeutics a slow and tedious process [8, 48, 18].

Nuclear Magnetic Resonance (NMR) spectroscopy is a cornerstone technique for small molecule structure elucidation. This experimental method provides information regarding the connectivity and local environment of, typically, each proton ($^1$H) and carbon ($^{13}$C) in a molecule, thus allowing the structure of a molecule to be deduced. However, the inverse problem of inferring the chemical structure from these spectral measurements is a challenging puzzle, which largely proceeds manually and requires significant time and expertise, even with computational assistance [8]. Consequently, automating molecular structure elucidation directly from raw 1D NMR spectra would significantly accelerate progress in chemistry, biomedicine, and natural product drug discovery [56, 47, 77, 42, 17].

With the rise of deep learning approaches applied to molecules, diffusion generative models [20, 59, 32] have emerged as powerful tools for tasks such as small molecule generation [21, 46, 69, 40], ligand-protein docking [10, 57], and protein structure prediction [2, 73] and design [25, 70, 15]. While early approaches emphasize 3D geometric symmetries via equivariant networks, recent trends suggest that non-equivariant transformers scale more effectively with model and data size and better capture 3D structures with data augmentation [69, 2].

In this work, we tackle the challenging task of NMR structure elucidation for complex natural products. We introduce CHEFNMR (CHemical Elucidation From NMR), an end-to-end diffusion model designed to infer an unknown molecule's structure from its 1D NMR spectra and chemical formula. CHEFNMR processes NMR spectra using a hybrid transformer with a convolutional tokenizer designed to capture multiscale spectral features, which are then used to condition a Diffusion Transformer [49] for 3D atomic structure generation. To scale to the complex chemical groups found in natural products, we curate SpectraNP, a large-scale dataset of synthetic 1D NMR spectra for 111,181 complex natural products (up to 274 atoms), significantly expanding the chemical complexity of prior datasets (≤101 atoms) [22, 4]. We compare CHEFNMR against chemical language model-based and graph-based formulations and demonstrate state-of-the-art accuracy across multiple synthetic and experimental benchmarks.

## 2 Background

NMR spectroscopy is a widely used analytical technique in chemistry for determining the structures of small molecules and biomolecules. A typical one-dimensional (1D) NMR experiment measures the response of all spin-active nuclei of a given type, for example hydrogen ($^1$H) or isotopic carbon ($^{13}$C), to radiofrequency pulses in a strong magnetic field. The resulting spectrum consists of peaks from chemically distinct nuclei, where peak positions (i.e., chemical shifts), intensities, and fine splitting patterns (i.e., $J$-coupling) reflect local chemical environments and connectivities of the nuclei.

Formally, let the observed spectrum be a real-valued signal $S(\delta) : \mathbb{R} \rightarrow \mathbb{R}$, where $\delta$ denotes the chemical shift (in parts per million, ppm) along the x-axis. The signal can be modeled as a sum over $N$ resonance peaks corresponding to each spin-active nucleus:

$$S(\delta) = \sum_{i=1}^{N} A_i \cdot L(\delta; \delta_i, \gamma_i) + \epsilon(\delta) \tag{1}$$

where $A_i$ is the intensity (amplitude) of the $i$-th peak, $\delta_i$ is the chemical shift (peak center), and $\gamma_i$ is the linewidth (related to relaxation) of the peak. $L(\delta; \delta_i, \gamma_i)$ is the normalized Lorentzian line shape:

$$L(\delta; \delta_i, \gamma_i) = \frac{1}{\pi} \cdot \frac{\gamma_i}{(\delta - \delta_i)^2 + \gamma_i^2} \tag{2}$$

and $\epsilon(\delta)$ models additive noise (e.g., Gaussian white noise or baseline drift). $J$-coupling refers to the splitting of the signal for a given nucleus into a sum of multiple peaks when nearby atoms interact:

$$S(\delta) = \sum_{i=1}^{N} \sum_{k=1}^{M_i} A_{ik} \cdot L(\delta; \delta_{ik}, \gamma_{ik}) + \epsilon(\delta) \tag{3}$$

where $M_i$ is the number of split components for the $i$-th nucleus, and $\delta_{ik}$ encodes the shifted peak positions. $J$-coupling occurs when other spin-active nuclei are within 2–4 edges in the molecular graph, and the signal splits into $M_i = m + 1$ components assuming $m$ interacting nuclei. See Figure 1 for an example.

Together, these features encode rich information about the types of chemical groups present and their connectivities, enabling chemists to deduce the underlying molecular structure. For example, certain chemical groups produce peaks that appear at an established range (e.g., aromatic ring-protons are detected at 6.5–8 ppm), whose exact location depends on the amount of chemical shielding from nearby atoms in a given molecule. These patterns, in addition to experimental noise due to the instrument, impurities, and solvent effects, make the inverse problem of deducing structure an extremely challenging task. NMR structure elucidation thus typically relies on additional information from 2D NMR experiments, prior information on the substructures present, or chemical formula from high-resolution mass spectrometry [7] combined with isotopic abundance and distribution patterns. In this work, we utilize the chemical formula as auxiliary input, as it is typically the most readily obtainable among common priors and effectively constrains the space of candidate molecular structures for complex natural products.

## 3 Method

In this section, we present CHEFNMR, an end-to-end diffusion model for molecular structure elucidation from 1D NMR spectra and the chemical formula. Our approach consists of two key components: NMR-ConvFormer for spectral embedding (Section 3.1) and a conditional diffusion model for 3D atomic coordinate generation (Section 3.2).

In CHEFNMR, we represent molecule-spectrum pairs as $(\boldsymbol{A}, \boldsymbol{X}, \mathcal{S})$, where $\boldsymbol{A} \in \{0, 1\}^{N \times d_{\text{atom}}}$ denotes the one-hot encoding of atom types for a molecule with $N$ atoms and $d_{\text{atom}}$ possible atom types, $\boldsymbol{X} \in \mathbb{R}^{N \times 3}$ represents the 3D atomic coordinates, and $\mathcal{S} = (\boldsymbol{s}_{\text{H}}, \boldsymbol{s}_{\text{C}})$ contains the NMR spectra, specifically the $^1$H spectrum $\boldsymbol{s}_{\text{H}} \in \mathbb{R}^{d_{\text{H}}}$ and the $^{13}$C spectrum $\boldsymbol{s}_{\text{C}} \in \mathbb{R}^{d_{\text{C}}}$. Our objective is to generate the 3D coordinates $\boldsymbol{X}$ conditioned on the atom types $\boldsymbol{A}$ (i.e., chemical formula) and the spectra $\mathcal{S}$ by sampling from the conditional probability distribution $p(\boldsymbol{X}|\boldsymbol{A}, \mathcal{S})$.

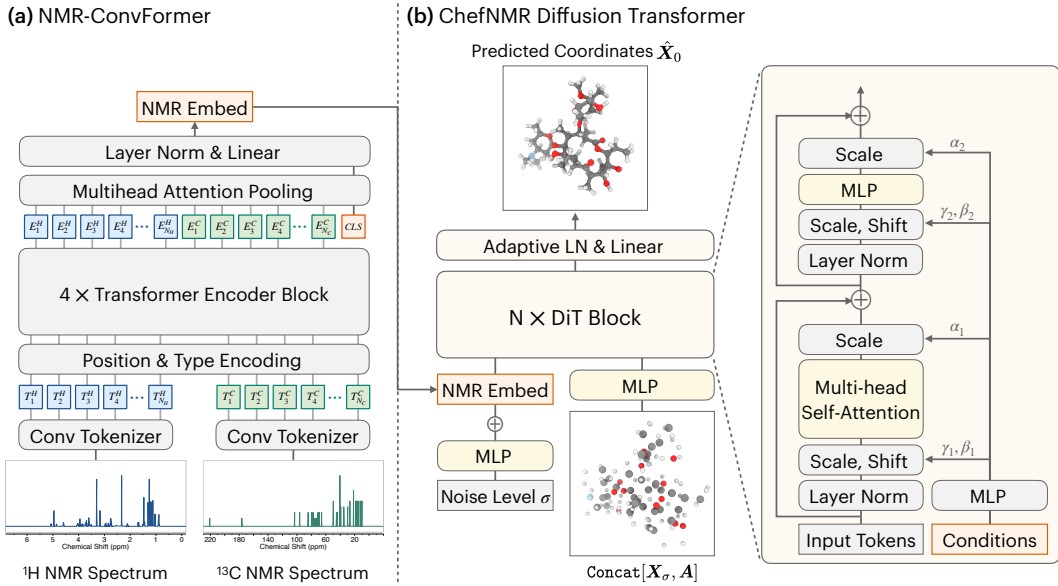

Figure 2: Overview of the CHEFNMR architecture. **(a)** NMR-ConvFormer processes 1D NMR spectra into a vector embedding using the convolutional tokenizer, transformer encoder, and multihead attention pooling (MAP). **(b)** Diffusion Transformer predicts clean 3D coordinates $\hat{X}_0$ from atom tokens formed by concatenating noisy coordinates $X_\sigma$ and atom types $A$, conditioned on the spectral embedding and noise level $\sigma$ via adaptive layer normalization [49].

## 3.1 NMR-ConvFormer: A Hybrid Convolutional Transformer for NMR spectral embedding

To effectively condition the generative process on the NMR spectra $\mathcal{S}$, we propose NMR-ConvFormer, an encoder designed to capture both local spectral features and global correlations within and between the $^1$H and $^{13}$C spectra, as shown in Figure 2(a). Unlike prior methods that rely solely on 1D convolutions [22, 45] or transformers with simple patching [67, 63], NMR-ConvFormer uses a hybrid approach, combining a convolutional tokenizer for local feature extraction and a transformer encoder for modeling complex intra- and inter-spectral dependencies.

**Convolutional Tokenizer.** Each input spectrum ($^1$H and $^{13}$C) is processed independently by a convolutional tokenizer comprising two 1D convolutional layers with ReLU and max-pooling, similar to [22]. This reduces sequence length while increasing channel dimensions, summarizing local patterns like peak intensity and splitting. The output is linearly projected to dimension $D_{\text{encoder}}$, yielding token sequences of shape $(T, D_{\text{encoder}})$.

**Transformer Encoder.** The token sequence, augmented with positional and type embeddings, is processed by a standard transformer encoder comprising multi-head self-attention and feed-forward networks with pre-layer norm and residuals. Self-attention captures patterns within each NMR spectrum and across different spectra, such as related peaks in a $^1$H spectrum or matching signals from the same chemical group in both $^1$H and $^{13}$C spectra.

**Multihead Attention Pooling (MAP).** We use MAP [37, 79] to obtain a fixed-size spectral embedding. A learnable [CLS] token prepended to the encoder output sequence aggregates information via a final self-attention layer. The resulting [CLS] token state, after layer normalization and linear projection, serves as the conditioning vector $z_{\mathcal{S}} \in \mathbb{R}^{D_{\text{hidden}}}$ for the diffusion model. Dropout is applied at multiple stages to mitigate overfitting. See Appendix D.2 for detailed hyperparameter settings.

## 3.2 Conditional 3D Atomic Diffusion Model

**Training Objective.** We adapt the EDM diffusion framework to conditional 3D molecular generation [32]. The model $D_\theta$ is trained to predict clean 3D coordinates $X_0$ from noisy inputs $X_\sigma = X_0 + n$, where $n \sim \mathcal{N}(0, \sigma^2 I)$ and the noise level $\sigma$ is sampled from a pre-defined

distribution $p(\sigma)$. Given $\boldsymbol{X}_\sigma$, $\sigma$, atom types $\boldsymbol{A}$, and spectral embedding $\boldsymbol{z}_\mathcal{S}$, the model minimizes:

$$\mathcal{L}_{\text{diffusion}} = \mathop{\mathbb{E}}_{\substack{(\boldsymbol{X}_0, \boldsymbol{A}, \boldsymbol{z}_\mathcal{S}) \sim p_{\text{data}}, \\ \sigma \sim p(\sigma), \boldsymbol{n} \sim \mathcal{N}(\boldsymbol{0}, \sigma^2 \boldsymbol{I})}} \left[ \lambda(\sigma) \mathcal{L}_{\text{MSE}}(\hat{\boldsymbol{X}}_0, \boldsymbol{X}_0) + \mathcal{L}_{\text{smooth\_lddt}}(\hat{\boldsymbol{X}}_0, \boldsymbol{X}_0) \right], \qquad (4)$$

where $\hat{\boldsymbol{X}}_0 = D_\theta(\boldsymbol{X}_\sigma; \sigma, \boldsymbol{A}, \boldsymbol{z}_\mathcal{S})$ are the predicted coordinates.

The MSE loss, $\mathcal{L}_{\text{MSE}} = \|\hat{\boldsymbol{X}}_0 - \boldsymbol{X}_0\|_2^2$, enforces global structure alignment. To ensure local geometric accuracy (e.g., bond lengths), crucial for chemical validity and often poorly captured by MSE alone, we add a smooth Local Distance Difference Test (LDDT) loss [43], adapted from AlphaFold3 [2]. The LDDT score is computed over all distinct atom pairs $(i, j)$:

$$\text{LDDT}(\hat{\boldsymbol{X}}_0, \boldsymbol{X}_0) = \frac{1}{N(N-1)} \sum_{i \neq j} \epsilon_{ij}, \quad \text{where} \quad \epsilon_{ij} = \frac{1}{4} \sum_{k=1}^{4} \text{sigmoid}(t_k - |\hat{d}_{ij} - d_{ij}|). \quad (5)$$

Here, $\hat{d}_{ij} = \|\hat{\boldsymbol{x}}_i - \hat{\boldsymbol{x}}_j\|_2$ and $d_{ij} = \|\boldsymbol{x}_{0,i} - \boldsymbol{x}_{0,j}\|_2$ are the predicted and true distances, respectively. Thresholds $t_k \in \{0.5, 1.0, 2.0, 4.0\,\text{Å}\}$ specify allowable deviations between predicted and true distances when evaluating prediction accuracy. The smooth LDDT loss is $\mathcal{L}_{\text{smooth\_lddt}} = 1 - \text{LDDT}$, encourages local geometric fidelity by penalizing pairwise deviations. The combined loss promotes both global alignment and local chemical validity.

**Random Coordinate Augmentation.** For each molecule, we generate $k$ ground-truth conformers. During training, we randomly sample one conformer $\boldsymbol{X}_0$ and apply a random rigid transformation (translation and rotation) following [2, 29, 40]. This augmentation encourages $D_\theta$ to learn $SE(3)$-invariant representations and mitigate overfitting, significantly improving performance.

**Diffusion Transformer (DiT) Architecture.** The network $D_\theta$ is a DiT [49] shown in Figure 2(b). Input atom tokens are formed by concatenating noisy coordinates $\boldsymbol{X}_\sigma$ and atom types $\boldsymbol{A}$, followed by an MLP projection. The noise level $\sigma$ is embedded using frequency encoding and an MLP. This noise embedding is added to the spectral embedding $\boldsymbol{z}_\mathcal{S}$ to form the conditioning vector, which is integrated into the DiT blocks via adaptive layer normalization (adaLN-Zero) [49].

**Conditional Dropout and Classifier-Free Guidance.** To improve robustness and flexibility in conditioning on different NMR spectra, we adopt classifier-free guidance (CFG) [19]. During training, the $^1$H NMR spectrum is dropped with probability $p_{\text{H}} = 0.1$, the $^{13}$C NMR spectrum is dropped with $p_{\text{C}} = 0.1$, and both are dropped simultaneously with $p_{\text{both}} = 0.1$. At inference, conditional and unconditional predictions are combined via

$$D_\theta^\omega(\boldsymbol{X}_\sigma; \sigma, \boldsymbol{A}, \boldsymbol{z}_\mathcal{S}) = (1 + \omega) \, D_\theta(\boldsymbol{X}_\sigma; \sigma, \boldsymbol{A}, \boldsymbol{z}_\mathcal{S}) - \omega \, D_\theta(\boldsymbol{X}_\sigma; \sigma, \boldsymbol{A}), \qquad (6)$$

where $\omega \geq 0$ controls guidance scale. This enables generation conditioned on either or both spectra, improving versatility and performance. See Appendix A and D.2 for full training and sampling algorithms and hyperparameter settings.

## 4 Related Work

**NMR Spectra Prediction.** The forward task of predicting a given molecule's NMR spectra is relatively established, facilitating data analysis and enabling the generation of simulatd datasets for structure elucidation of simple compounds via database retrieval. These spectra prediction methods range from precise, computationally intensive quantum-chemical simulations to more recent exploratory ML approaches [6, 28, 13, 35, 31, 16, 39, 44]. Following established dataset curation practices [22, 4], we create our `SpectraNP` dataset using the commercial software MestReNova [44], which combines closed-source ML and chemoinformatics algorithms.

**NMR Structure Elucidation.** Structure elucidation from NMR spectroscopy is a challenging inverse problem, due to the complexity of spectra data and the vast chemical space [60, 17]. Traditional computer-aided systems, while historically employed and useful, often suffer from computational inefficiencies [8]. Recent ML methods have tackled this challenge, but most simplify the problem by predicting molecular substructures instead of full molecules [38, 36, 5, 64, 76, 33], or by leveraging richer inputs, such as multimodal spectra beyond NMR [54, 12, 50, 55, 11, 9, 63, 53] and database retrieval [78, 62, 33]. In contrast, our method directly tackles the *de novo* elucidation of molecular structures using only raw 1D NMR spectra and chemical formulae.

***De Novo* Structure Elucidation from 1D NMR Spectra.** Recent work developing machine learning methods for *de novo* structure elucidation from 1D NMR spectra focuses on structurally simple molecules, leveraging either chemical language models or graph-based models. Chemical language models [80, 22, 3, 4] generate SMILES strings [71], a sequence-based molecular representation. For example, Hu et al. [22] use a multitask transformer pre-trained on 3.1M substructure-molecule pairs and fine-tuned on 143k NMR spectra from `SpectraBase` [26], achieving 69.6% top-15 accuracy for molecules under 59 atoms. Alberts et al. [3, 4] employ transformers to predict SMILES from text-based 1D NMR peak lists and chemical formulas, reporting 89.98% top-10 accuracy on the `USPTO` dataset [41] for molecules under 101 atoms. Graph-based models iteratively construct molecular graphs with GNNs, using methods like Markov decision processes or Monte Carlo tree search [27, 23, 61]. However, these methods do not handle molecules with more than 64 atoms or large rings (>8 atoms), likely due to the limited availability of large-scale spectral datasets and the high computational cost of search-based algorithms for complex molecules. To the best of our knowledge, CHEFNMR is the first method based on 3D atomic diffusion models for NMR structure elucidation that scales to complex natural products.

**3D Molecular Diffusion Models.** Diffusion models have emerged as powerful tools for 3D molecular generation. E(3)-equivariant GNNs [74, 21, 75, 46] enforce geometric constraints, but non-equivariant transformers are increasingly favored for their scalability and performance in small molecule generation [69, 40, 29] and protein structure prediction [2, 15] involving hundreds of thousands of atoms. Inspired by these recent trends, we apply a scalable DiT [49] to generate 3D atomic structures from NMR spectra, exploiting their scalability and expressivity by creating a large synthetic NMR spectra dataset of natural products.

## 5 Experiments

### 5.1 Dataset Curation

**Synthetic Datasets.** We evaluate models on two public benchmarks, `SpectraBase` [22] and `USPTO` [4], and our self-curated `SpectraNP` dataset. `SpectraBase` contains simple molecules [22, 26], while `USPTO` features a broader range of molecules in chemical reactions [41]. `SpectraNP` combines data from NPAtlas [52], a database of small molecules from bacteria and fungi, with a subset of NP-MRD [72] including various natural products.

Table 1: Summary of dataset statistics.

| Synthetic | # Molecules | # Atoms |
|---|---|---|
| SpectraBase [22] | 141k | [3, 59] |
| USPTO [4] | 745k | [8, 101] |
| SpectraNP | 111k | [4, 274] |
| **Experimental** | **# Molecules** | **# Solvents** |
| SpecTeach [65] | 238 | 2 |
| NMRShiftDB2 [34] | 23k | >7 |

**Experimental Datasets.** To evaluate the ability of models trained on synthetic data to generalize to experimental data, we curate two experimental datasets. Following [4], we include the `SpecTeach` dataset [65], which contains 238 simple molecules for spectroscopy education. We also include `NMRShiftDB2` [34], a larger-scale dataset of $^{13}$C NMR spectra in various solvents, following [61, 28]. These experimental datasets include impurities, solvents, and baseline noise (See Figure 5), enabling robustness testing for experimental variations.

**Data Structure and Preprocessing.** Each data entry is a tuple *(SMILES, $^1$H NMR spectrum, $^{13}$C NMR spectrum, atom features)*. SMILES strings are canonicalized with stereochemistry removed, and synthetic spectra are simulated using MestreNova [44]. Atom features include atom types $A$ and 3D conformations $X$, generated using RDKit's `ETKDGv3` algorithm [1] given the SMILES string.

To preprocess datasets, any duplicate SMILES are first removed. $^1$H and $^{13}$C spectra are interpolated to 10,000-dimensional vectors following [22, 4], and normalized by their highest peak intensity, except for `SpectraBase` [22] and experimental datasets, where $^{13}$C spectra are binned into 80 binary vectors. To validate 3D conformers, SMILES strings are reconstructed from atom types and 3D coordinates using RDKit's `DetermineBonds` function [1], and molecules failing reconstruction are discarded. See Appendix C for dataset curation details.

### 5.2 Experimental Setup

**Baselines.** We compare CHEFNMR against two existing chemical language models and introduce a graph-based model to assess the impact of molecular representations on the structure elucidation task.

The chemical language models are: (1) **Hu et al.** [22] propose a two-stage multitask transformer for predicting SMILES from 1D NMR spectra. Their method pre-defines 957 substructures and pre-trains a substructure-to-SMILES model on 3.1M molecules, and then fine-tunes a multitask transformer on 143k NMR spectra from `SpectraBase`. We retrain their substructure-to-SMILES model on the same 3.1M dataset and fine-tune it on each synthetic benchmark. (2) **Alberts et al.** [4] develop a transformer to predict stereochemical SMILES from text-based 1D NMR peak lists and chemical formulae. Due to unavailable inference code and differences in input (peak lists vs. raw spectra) and output (stereo vs. non-stereo SMILES), we report their published results on USPTO and `SpecTeach`.

To test an alternative graph-based representation, we also propose **NMR-DiGress**, a model integrating the discrete graph diffusion model DiGress [66] with our NMR-ConvFormer. Molecular graphs are represented as atom types $A$ and bond matrices $E \in \{0, 1\}^{N \times N \times d_{\text{bond}}}$, where $N$ is the number of atoms and $d_{\text{bond}}$ is the number of bond types. DiGress adds noise to each atom or bond independently via discrete Markov chains, and trains a graph transformer to reverse this process to generate molecular graphs. We adapt DiGress to condition on spectral features from NMR-ConvFormer and atom types $A$, generating only bond matrices. See Appendix B and D.1 for full algorithms and detailed settings.

**CHEFNMR.** We evaluate two variants: CHEFNMR-S (134M parameters) and CHEFNMR-L (462M parameters) with the same NMR-ConvFormer and different sizes of DiT. See Appendix D.2 for additional experimental details.

**Metrics.** We evaluate models using: (1) **Top-$k$ matching accuracy**, which checks whether the ground truth SMILES string is exactly matched by any of the top-$k$ predicted molecules. For non-language models, we reconstruct canonical, non-stereo SMILES from the predicted molecular graph (atom types and generated bond matrix) or 3D structure (atom types and generated 3D coordinates) using RDKit [1]. (2) **Top-$k$ maximum Tanimoto similarity**, which evaluates structural similarity between the ground truth and the most similar molecule in the top-$k$ predictions, using the Tanimoto similarity of Morgan fingerprints (length 2048, radius 2) [1].

# 6 Results

This section presents the quantitative and qualitative results across benchmarks. Section 6.1 shows CHEFNMR's state-of-the-art performance on synthetic datasets, and Section 6.2 demonstrates robust zero-shot generalization on experimental datasets. Section 6.3 presents ablation studies on the contributions of the diffusion training process and the NMR-ConvFormer spectra embedder.

## 6.1 Performance on Synthetic Spectra

Table 2 summarizes the performance on synthetic $^1$H and $^{13}$C NMR spectra. CHEFNMR significantly surpasses all baselines in matching accuracy and maximum Tanimoto similarity across datasets. The

Table 2: Performance on synthetic $^1$H and $^{13}$C NMR spectra, reported as the mean $\pm$ standard deviation over three independent sampling runs. Acc%: accuracy; Sim: Tanimoto similarity. *: reported results. N/A: not applicable.

| Dataset | Model | Top-1 | | Top-5 | | Top-10 | |
|---|---|---|---|---|---|---|---|
| | | Acc% ↑ | Sim ↑ | Acc% ↑ | Sim ↑ | Acc% ↑ | Sim ↑ |
| SpectraBase | Hu et al. | 45.24±.18 | 0.686±.001 | 62.37±.08 | 0.815±.001 | 67.38±.05 | 0.847±.001 |
| | NMR-DiGress | 43.56±.30 | 0.625±.002 | 62.47±.20 | 0.779±.002 | 68.39±.35 | 0.817±.001 |
| | CHEFNMR-S | 69.15±.08 | 0.807±.002 | 82.09±.24 | 0.904±.002 | 85.30±.04 | 0.922±.000 |
| | CHEFNMR-L | **72.04±.02** | **0.833±.000** | **85.24±.10** | **0.923±.001** | **88.20±.07** | **0.940±.000** |
| USPTO | Hu et al. | 38.02±.02 | 0.674±.001 | 55.85±.04 | 0.810±.000 | 61.76±.03 | 0.845±.000 |
| | Alberts et al.* | 73.38±.08 | N/A | 87.94±.14 | N/A | 89.98±.16 | N/A |
| | NMR-DiGress | 22.51±.13 | 0.504±.000 | 41.26±.12 | 0.708±.001 | 48.87±.11 | 0.761±.000 |
| | CHEFNMR-S | 81.16±.08 | 0.902±.000 | 91.03±.05 | 0.964±.000 | 92.90±.05 | **0.973±.000** |
| | CHEFNMR-L | **81.57±.09** | **0.912±.000** | **91.09±.11** | **0.965±.000** | **93.01±.05** | **0.973±.000** |
| SpectraNP | Hu et al. | 19.26±.10 | 0.585±.001 | 34.00±.19 | 0.736±.001 | 39.87±.02 | 0.774±.001 |
| | NMR-DiGress | 2.12±.14 | 0.260±.001 | 6.31±.08 | 0.432±.001 | 9.17±.11 | 0.485±.000 |
| | CHEFNMR-S | **40.37±.33** | 0.583±.004 | 59.08±.28 | 0.791±.000 | 64.37±.08 | 0.834±.001 |
| | CHEFNMR-L | 40.15±.29 | **0.631±.004** | **59.83±.30** | **0.822±.002** | **65.74±.09** | **0.860±.000** |

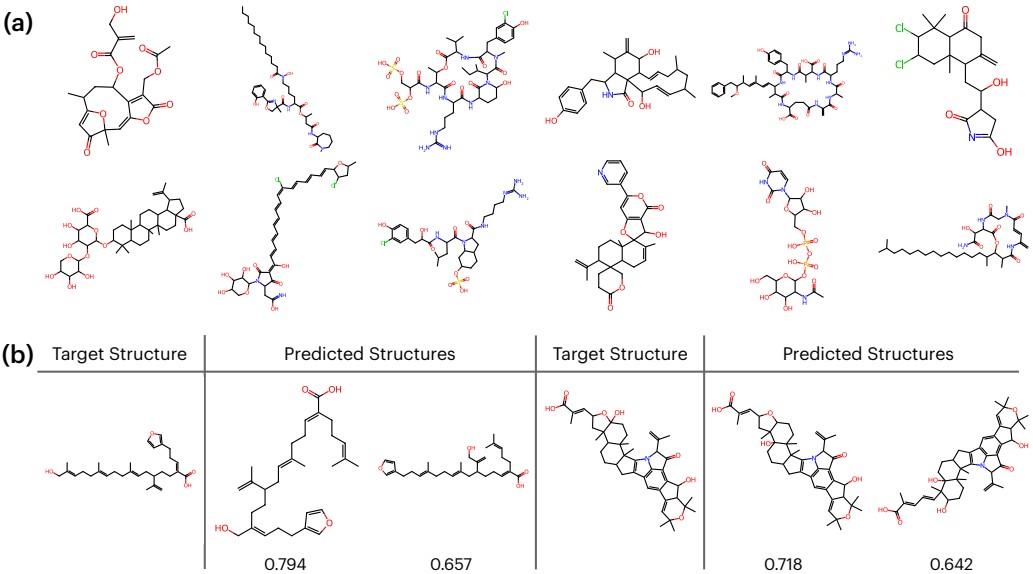

**(b)**

| Target Structure | Predicted Structures | | Target Structure | Predicted Structures | |
|---|---|---|---|---|---|
| | 0.794 | 0.657 | | 0.718 | 0.642 |

Figure 3: Examples of CHEFNMR's predictions on the synthetic `SpectraNP` dataset. **(a)** Correctly predicted diverse and complex natural products in top-1 predictions. **(b)** Incorrect top-2 predictions ranked by Tanimoto similarity remain chemically valid and structurally similar to the ground truth.

advantage is most pronounced on the challenging `SpectraNP` dataset, where CHEFNMR achieves 40% top-1 accuracy compared to 19% for Hu et al. and only 2% for NMR-DiGress.

Performance scales up with both model and dataset size. CHEFNMR-S outperforms baselines by large margins across all datasets, and CHEFNMR-L further improves accuracy. Larger datasets also yield better results, with the highest performance observed on `USPTO` (745k data), followed by `SpectraBase` (141k data) and `SpectraNP` (111k data). This suggests expanding `SpectraNP` could further enhance performance in elucidating complex natural products.

Figure 3 provides qualitative examples of CHEFNMR's performance on `SpectraNP`. CHEFNMR accurately predicts diverse and complex natural product structures in its top-1 predictions (Figure 3(a)). We additionally show incorrect predictions (Figure 3(b)), and find that many of the generated structures remain chemically valid and similar to the ground truth. Additional qualitative examples are provided in Appendix F.4.

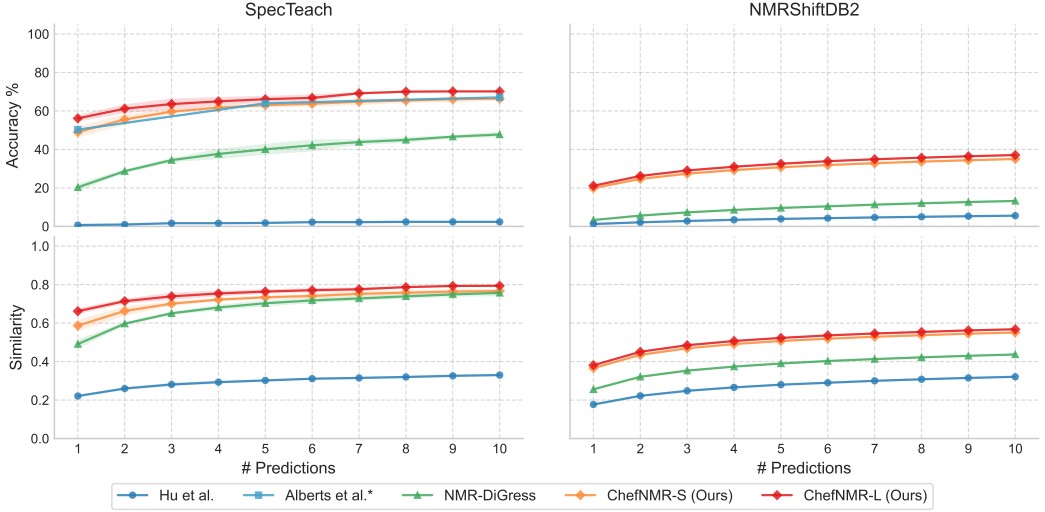

Figure 4: Zero-shot performance on experimental NMR spectra, shown as the mean $\pm$ standard deviation over three independent sampling runs. Models are trained on `USPTO`. Evaluation is on $^1$H and $^{13}$C spectra for `SpecTeach`, and on $^{13}$C spectra for `NMRShiftDB2`.*: reported results.

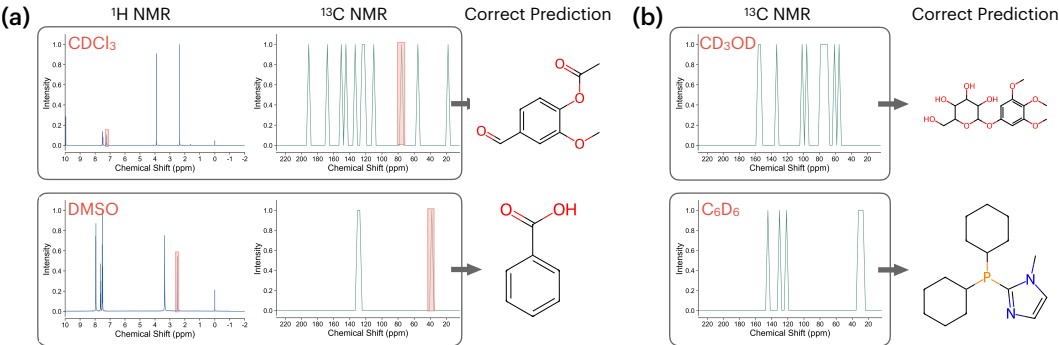

Figure 5: Examples of correct structures in CHEFNMR's top-1 predictions on experimental **(a)** `SpecTeach` and **(b)** `NMRShiftDB2` datasets respectively, with solvent peaks marked in red.

## 6.2 Zero-shot Performance on Experimental Spectra

Figure 4 reports the zero-shot performance on experimental $^1$H and $^{13}$C NMR spectra. CHEFNMR achieves 56% top-1 accuracy on `SpecTeach` and 21% on `NMRShiftDB2`, significantly outperforming Hu et al. [22] and NMR-DiGress, which generalize poorly to both experimental benchmarks. Figure 5 shows that CHEFNMR can generate the correct structures in its top-1 predictions despite substantial experimental variations, such as solvent effect and impurities.

## 6.3 Ablation Studies

We perform extensive ablation studies to assess the contributions of key components in diffusion training and the NMR-ConvFormer on the `SpectraBase` dataset. Each row in Table 3 corresponds to a variant with one component removed or modified from the base setting. Coordinate augmentation improves accuracy by 20%, indicating learning symmetries is crucial. Within the NMR-ConvFormer, convolutional tokenizer, MAP pooling, and dropout are relatively important. Detailed settings and additional ablations, including separate contributions of $^1$H and $^{13}$C spectra, are provided in Appendix E.

Table 3: Ablation results (Top-1 Acc% / Sim).

| Configuration | Acc@1% ↑ | Sim@1 ↑ |
|---|---|---|
| Base (CHEFNMR-S) | **69.15** | **0.807** |
| *Diffusion Training Ablation* | | |
| – Coord Augmentation | 49.75 | 0.651 |
| – Smooth LDDT Loss | 68.31 | 0.798 |
| *NMR-ConvFormer Ablation* | | |
| – Conv Tokenizer | 61.78 | 0.756 |
| – Token Count Reduction | 66.28 | 0.789 |
| – Transformer Block | 68.12 | 0.797 |
| – MAP Pooling | 62.97 | 0.758 |
| – Dropout | 65.48 | 0.777 |

## 7 Conclusion

In this work, we address the challenge of determining structures for complex natural products directly from raw 1D NMR spectra and chemical formulas. We introduce CHEFNMR, an end-to-end diffusion model that combines a hybrid convolutional transformer for spectral encoding with a Diffusion Transformer for 3D molecular structure generation. To encompass the chemical diversity present in natural products, we curate `SpectraNP`, a large-scale dataset of synthetic 1D NMR spectra for natural products. Our approach achieves state-of-the-art accuracy on synthetic and experimental benchmarks, with ablation studies validating the importance of key design components.

Several limitations highlight promising directions for future work. Expanding the training set to include experimental spectra and more natural products could further improve model performance. Additional information, such as 2D NMR spectra, could be incorporated to resolve stereochemistry. Furthermore, adding a confidence module could help chemists better assess the reliability of predicted structures. Overall, automating NMR-based structure elucidation has the potential to significantly accelerate molecular discovery. Careful validation and responsible deployment will be essential to ensure safe and impactful use in real-world applications.

## Acknowledgements

The authors acknowledge the use of computing resources at Princeton Research Computing, a consortium of groups led by the Princeton Institute for Computational Science and Engineering (PICSciE) and Office of Information Technology's Research Computing. The Zhong lab is grateful for support from the Princeton Catalysis Initiative, Princeton School of Engineering and Applied Sciences, Chan Zuckerberg Imaging Institute, Janssen Pharmaceuticals, and Generate Biomedicines. The Seyedsayamdost lab is grateful for support from the Princeton Catalysis Initiative and a Princeton SEAS grant. The funders had no role in study design, data collection and analysis, decision to publish or preparation of the manuscript. The authors also thank the silhouette of *Actinomyces* used in Figure 1 created by Matt Crook and licensed under CC BY-SA 3.0 Unported.

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

# A    Details of Conditional 3D Atomic Diffusion Model

In this section, we provide full training and sampling algorithms for the conditional 3D atomic diffusion model described in Section 3.2 of the main paper.

**Training Procedure.** The complete training procedure is outlined in Algorithm 1. The smooth LDDT loss is detailed in Algorithm 2, adapted from AlphaFold3 [2]. Unlike the original, which computes the smooth LDDT loss within a certain radius for proteins [2], we compute it of all atom pairs for small molecules, as small molecules are more compact in 3D space than proteins.

---

**Algorithm 1** Diffusion Training.

---

1: **procedure** TRAINDIFFUSION($D_\theta$, atom types $\boldsymbol{A}$, ground-truth conformers $\{\boldsymbol{X}^*\}_{k=1}^{K=3}$, NMR spectra $\mathcal{S} = (\boldsymbol{s}_\text{H}, \boldsymbol{s}_\text{C})$, noise schedule $(P_\text{mean}, P_\text{std}) = (-1.2, 1.3)$, standard deviation of atom coordinates $\sigma_\text{data}$ )
2:      $\mathcal{S} \leftarrow (\boldsymbol{s}_\text{H}, \boldsymbol{0})$, $(\boldsymbol{0}, \boldsymbol{s}_\text{C})$, or $(\boldsymbol{0}, \boldsymbol{0})$ with probability 0.1 each          ▷ Randomly drop spectra
3:      $\boldsymbol{z}_\mathcal{S} \leftarrow$ NMR-CONVFORMER($\mathcal{S}$)                                                   ▷ Encode spectra
4:      sample $k \sim$ Uniform($\{1, \dots, K\}$); $\boldsymbol{X}_0 \leftarrow \boldsymbol{X}_k^*$             ▷ Select a target conformer
5:      $\boldsymbol{X}_0 \leftarrow \boldsymbol{X}_0 - \bar{\boldsymbol{X}}_0$                                        ▷ Center coordinates
6:      sample $\boldsymbol{R} \sim$ SO(3), $\boldsymbol{t} \sim \mathcal{N}(\boldsymbol{0}, \boldsymbol{I})$; $\boldsymbol{X}_0 \leftarrow \boldsymbol{R}\boldsymbol{X}_0 + \boldsymbol{t}$          ▷ Random rigid transformation
7:      sample $\ln \sigma \sim \mathcal{N}(P_\text{mean}, P_\text{std}^2)$; $\sigma \leftarrow \sigma \cdot \sigma_\text{data}$                 ▷ Sample noise scale
8:      sample $\boldsymbol{n} \sim \mathcal{N}(\boldsymbol{0}, \sigma^2 \boldsymbol{I})$; $\boldsymbol{X}_\sigma \leftarrow \boldsymbol{X}_0 + \boldsymbol{n}$                    ▷ Add Gaussian noise
9:      $\hat{\boldsymbol{X}}_0 \leftarrow D_\theta(\boldsymbol{X}_\sigma; \sigma, \boldsymbol{A}, \boldsymbol{z}_\mathcal{S})$                                           ▷ Predict clean coordinates
10:     minimize $\mathcal{L}_\text{diffusion} = \lambda(\sigma)\|\hat{\boldsymbol{X}}_0 - \boldsymbol{X}_0\|_2^2 + \mathcal{L}_\text{smooth-lddt}(\hat{\boldsymbol{X}}_0, \boldsymbol{X}_0)$
11: **end procedure**

---

---

**Algorithm 2** Smooth LDDT Loss.

---

1: **procedure** SMOOTHLDDTLOSS(predicted coordinates $\hat{\boldsymbol{X}}_0$, ground-truth coordinates $\boldsymbol{X}_0$, thresholds $t = \{0.5, 1.0, 2.0, 4.0\}$)
2:      $\hat{d}_{ij} \leftarrow \|\hat{\boldsymbol{x}}_i - \hat{\boldsymbol{x}}_j\|_2$
3:      $d_{ij} \leftarrow \|\boldsymbol{x}_{0,i} - \boldsymbol{x}_{0,j}\|_2$                                             ▷ Compute pairwise distances
4:      $\Delta d_{ij} \leftarrow |\hat{d}_{ij} - d_{ij}|$                                                    ▷ Distance differences
5:      $\epsilon_{ij} \leftarrow \frac{1}{4}\sum_{k=1}^4 \text{sigmoid}(t_k - \Delta d_{ij})$               ▷ Preserved scores
6:      LDDT $\leftarrow \text{mean}_{i \neq j}(\epsilon_{ij})$                                   ▷ Mean score, excluding self-pairs
7:      $\mathcal{L}_\text{smooth\_lddt} \leftarrow 1 - $ LDDT                                           ▷ Smooth LDDT loss
8:      **return** $\mathcal{L}_\text{smooth\_lddt}$
9: **end procedure**

---

**Preconditioning.** To stabilize training across different noise levels, we precondition the denoising network $D_\theta$ following EDM [32]:

$$D_\theta(\boldsymbol{X}_\sigma; \sigma, \boldsymbol{A}, \boldsymbol{z}_\mathcal{S}) = c_\text{skip}(\sigma)\boldsymbol{X}_\sigma + c_\text{out}(\sigma)F_\theta\big(c_\text{in}(\sigma)\boldsymbol{X}_\sigma; c_\text{noise}(\sigma), \boldsymbol{A}, \boldsymbol{z}_\mathcal{S}\big), \qquad (7)$$

where $F_\theta$ is the core neural network performing the actual computation. The scaling functions $c_\text{skip}, c_\text{out}, c_\text{in}, c_\text{noise}$, and the loss weight $\lambda(\sigma)$ are defined as EDM [32]:

$$c_\text{skip}(\sigma) = \frac{\sigma_\text{data}^2}{\sigma^2 + \sigma_\text{data}^2}, \quad c_\text{out}(\sigma) = \frac{\sigma \cdot \sigma_\text{data}}{\sqrt{\sigma_\text{data}^2 + \sigma^2}}, \qquad (8)$$

$$c_\text{in}(\sigma) = \frac{1}{\sqrt{\sigma^2 + \sigma_\text{data}^2}}, \quad c_\text{noise}(\sigma) = \frac{1}{4}\ln(\sigma), \quad \lambda(\sigma) = \frac{\sigma^2 + \sigma_\text{data}^2}{(\sigma \cdot \sigma_\text{data})^2}. \qquad (9)$$

Here, $\sigma_\text{data}$ represents the standard deviation of atom coordinates in the dataset (see Appendix Table 7).

**Conditional Dropout and Classifier-Free Guidance.** To improve robustness and flexibility in conditioning on NMR spectra, we adopt classifier-free guidance (CFG) [19]. During training, the ${}^1$H NMR spectrum is replaced with zeros with probability $p_\text{H} = 0.1$, the ${}^{13}$C NMR spectrum is dropped with $p_\text{C} = 0.1$, and both are dropped simultaneously with $p_\text{both} = 0.1$ (see Algorithm 1). At inference, conditional and unconditional predictions are combined via

$$D_\theta^\omega(\boldsymbol{X}_\sigma; \sigma, \boldsymbol{A}, \boldsymbol{z}_\mathcal{S}) = (1 + \omega)D_\theta(\boldsymbol{X}_\sigma; \sigma, \boldsymbol{A}, \boldsymbol{z}_\mathcal{S}) - \omega D_\theta(\boldsymbol{X}_\sigma; \sigma, \boldsymbol{A}), \qquad (10)$$

where $\omega \geq 0$ controls guidance scale. This enables generation conditioned on either or both spectra, improving versatility and performance. In this paper, we set $\omega \in \{1, 1.5, 2\}$ depending on datasets.

---

**Algorithm 3** Diffusion Sampling using Stochastic Heun's 2^nd order Method.

---

1: **procedure** SAMPLEDIFFUSION($D_\theta^\omega(X_\sigma; \sigma, A, z_S)$, atom type $A$, NMR spectra embedding $z_S$, noise level schedule $\sigma_{i \in \{0,...,N\}}, \gamma_0 = 0.8, \gamma_{\min} = 1.0, \omega$ guidance scale)
2:   **sample** $X_0 \sim \mathcal{N}(0, \sigma_0^2 I)$
3:   **for** $i \in \{0, \ldots, N-1\}$ **do**
4:     $\gamma = \gamma_0$ if $\sigma_i > \gamma_{\min}$ else 0
5:     $\hat{\sigma}_i \leftarrow \sigma_i + \gamma\sigma_i$                                       ▷ Temporarily increase noise level $\hat{\sigma}_i$
6:     **sample** $\epsilon_i \sim \mathcal{N}(0, I)$
7:     $\hat{X}_i \leftarrow X_i + \sqrt{\hat{\sigma}_i^2 - \sigma_i^2}\, \epsilon_i$                    ▷ Add new noise to move from $\sigma_i$ to $\hat{\sigma}_i$
8:     $d_i \leftarrow (\hat{X}_i - D_\theta^\omega(\hat{X}_i; \hat{\sigma}_i, A, z_S))/\hat{\sigma}_i$             ▷ Evaluate $dX/d\sigma$ at $\hat{\sigma}_i$
9:     $X_{i+1} \leftarrow \hat{X}_i + (\sigma_{i+1} - \hat{\sigma}_i)d_i$                ▷ Take Euler step from $\hat{\sigma}_i$ to $\sigma_{i+1}$
10:     **if** $\sigma_{i+1} \neq 0$ **then**
11:       $d_i' \leftarrow (X_{i+1} - D_\theta^\omega(X_{i+1}; \sigma_{i+1}))/\sigma_{i+1}$          ▷ Apply 2^nd order correction
12:       $X_{i+1} \leftarrow \hat{X}_i + (\sigma_{i+1} - \hat{\sigma}_i)(\frac{1}{2}d_i + \frac{1}{2}d_i')$
13:     **end if**
14:   **end for**
15:   **return** $X_N$
16: **end procedure**

---

**Sampling Procedure.** The reverse diffusion process begins with $X_0 \sim \mathcal{N}(0, \sigma_{\max}^2 I)$ and iteratively denoises to obtain $X_N$. This process is governed by the stochastic differential equation (SDE) [32]:

$$\mathrm{d}X = \underbrace{-\sigma\nabla_X \log p(X; \sigma | A, z_S)\,\mathrm{d}\sigma}_{\text{probability flow ODE}} \underbrace{- \beta(\sigma)\sigma^2\nabla_X \log p(X; \sigma | A, z_S)\,\mathrm{d}\sigma + \sqrt{2\beta(\sigma)}\sigma\,\mathrm{d}w}_{\text{Langevin diffusion SDE}},$$

(11)

where $\nabla_X \log p(X; \sigma | A, z_S) = (D_\theta^\omega(X_\sigma; \sigma, A, z_S) - X)/\sigma^2$ is the conditional score function [24], $\sigma$ is the noise level, and $\mathrm{d}w$ is the Wiener process. The term $\beta(\sigma)$ determines the rate at which existing noise is replaced by new noise.

During inference, the noise level schedule $\sigma_{i \in \{0,...,N\}}$ is defined as EDM [32]:

$$\sigma_{i<N} = \left(\sigma_{\max}^{\frac{1}{\rho}} + \frac{i}{N-1}(\sigma_{\min}^{\frac{1}{\rho}} - \sigma_{\max}^{\frac{1}{\rho}})\right)^\rho, \quad \sigma_N = 0,$$

(12)

where $\sigma_{\max} = 80$, $\sigma_{\min} = 0.0004$, $\rho = 7$, and $N = 50$ is the number of diffusion steps. The sampling process is performed by solving the SDE using the stochastic Heun's 2^nd method [32], as outlined in Algorithm 3.

## B   Details of NMR-DiGress

As introduced in Section 5.2 of the main paper, NMR-DiGress is a graph-based baseline model integrating the discrete graph diffusion model DiGress [66] with our NMR-ConvFormer for molecular structure elucidation from 1D NMR spectra and the chemical formula. In this section, we provide a detailed description of the training and sampling procedures of NMR-DiGress.

In NMR-DiGress, molecule-spectrum pairs are represented as $(\mathcal{G}, \mathcal{S})$, where $\mathcal{G} = (A, E)$ is the molecular graph. The atom types $A \in \{0, 1\}^{N \times d_{\text{atom}}}$ and bond types $E \in \{0, 1\}^{N \times N \times d_{\text{bond}}}$ represent the graph, with $N$ being the number of heavy atoms (excluding hydrogens), and $d_{\text{atom}}$ and $d_{\text{bond}}$ being the total number of atom and bond types, respectively. Bond types include no bond, single bond, double bond, triple bond, and aromatic bond.

The objective of NMR-DiGress is to generate the bond types $E$ conditioned on the atom types $A$ (i.e., chemical formula) and the spectra $\mathcal{S}$ by sampling from the conditional probability distribution $p(E|A, \mathcal{S})$. Key differences from the original DiGress are: (1) Atom types are already known in NMR-DiGress, so only bond matrices are predicted. (2) Spectra embeddings $z_S$ from NMR-ConvFormer are added as graph-level features during training and sampling.

---

**Algorithm 4** NMR-DiGress Training.

---

1: **procedure** TRAIN NMR-DIGRESS(molecular graph $\mathcal{G} = (\boldsymbol{A}, \boldsymbol{E})$, NMR spectra embedding $\boldsymbol{z}_{\mathcal{S}}$)
2:   **sample** $t \sim \mathcal{U}[1, T]$  ▷ Sample a diffusion time from the uniform distribution
3:   **sample** $\boldsymbol{E}^t \sim \boldsymbol{E}\,\bar{\boldsymbol{Q}}^t$  ▷ Add noise to the bond matrix
4:   $\mathcal{G}^t \leftarrow (\boldsymbol{A}, \boldsymbol{E}^t)$
5:   $\boldsymbol{z}_{\mathcal{G}} \leftarrow f(\mathcal{G}^t, t)$  ▷ Structural features computed from the graph
6:   $\hat{p}^E \leftarrow \phi_\theta(\mathcal{G}^t, \boldsymbol{z}_{\mathcal{G}}, \boldsymbol{z}_{\mathcal{S}})$  ▷ Predict the bond matrix
7:   **minimize** $\ell_{CE}(\hat{p}^E, \boldsymbol{E})$  ▷ Cross-entropy loss
8: **end procedure**

---

**Algorithm 5** NMR-DiGress Sampling.

---

1: **procedure** SAMPLE NMR-DIGRESS(NMR spectra embedding $\boldsymbol{z}_{\mathcal{S}}$, atom types $\boldsymbol{A}$, marginal distribution of bond types $\boldsymbol{m}$, number of diffusion steps $T$)
2:   **sample** $\boldsymbol{E}^T \sim \boldsymbol{m}$  ▷ Independently sample each initial bond from the marginal distribution
3:   $\mathcal{G}^T \leftarrow (\boldsymbol{A}, \boldsymbol{E}^T)$
4:   **for** $t = T, \ldots, 1$ **do**
5:     $\boldsymbol{z}_{\mathcal{G}} \leftarrow f(\mathcal{G}^t, t)$  ▷ Structural features computed from the graph
6:     $\hat{p}^E \leftarrow \phi_\theta(\mathcal{G}^t, \boldsymbol{z}_{\mathcal{G}}, \boldsymbol{z}_{\mathcal{S}})$  ▷ Predict the bond matrix
7:     $p_\theta(e_{ij}^{t-1} \mid \mathcal{G}^t) \leftarrow \sum_e q(e_{ij}^{t-1} \mid e_{ij} = e, e_{ij}^t)\,\hat{p}_{ij}^E(e)$  ▷ Posterior distribution of each bond
8:     $\mathcal{G}^{t-1} \sim \boldsymbol{A} \times \prod_{ij} p_\theta(e_{ij}^{t-1} \mid \mathcal{G}^t)$  ▷ Reverse process
9:   **end for**
10:   **return** $\mathcal{G}^0$
11: **end procedure**

---

**Training Procedure.** The training procedure of NMR-DiGress is adapted from DiGress [66]. Noise is added to each bond independently via discrete Markov chains, and a neural network is trained to reverse this process to generate bond matrices.

Specifically, to add noise to a graph, we define a discrete Markov process $\{\boldsymbol{E}^t\}_{t=0}^T$ starting from the bond matrix $\boldsymbol{E}^0 = \boldsymbol{E}$:

$$q\left(\boldsymbol{E}^t | \boldsymbol{E}^{t-1}\right) = \boldsymbol{E}^{t-1}\boldsymbol{Q}^t, \tag{13}$$

where $\boldsymbol{Q}^t$ is the transition matrix from step $t-1$ to $t$. From the properties of the Markov chain, the distribution of $\boldsymbol{E}^t$ given $\boldsymbol{E}$ is:

$$q\left(\boldsymbol{E}^t | \boldsymbol{E}\right) = \boldsymbol{E}\bar{\boldsymbol{Q}}^t, \tag{14}$$

where $\bar{\boldsymbol{Q}}^t = \boldsymbol{Q}^1\boldsymbol{Q}^2...\boldsymbol{Q}^t$. Following DiGress, we use the noise schedule:

$$\bar{\boldsymbol{Q}}^t = \bar{\alpha}^t \boldsymbol{I} + \bar{\beta}^t \boldsymbol{1}\,\boldsymbol{m}^\top, \tag{15}$$

where $\bar{\alpha}^t = \cos\left(0.5\pi\left(t/T + s\right)/\left(1 + s\right)\right)^2$, $\bar{\beta}^t = 1 - \bar{\alpha}^t$, $T = 500$ is the number of diffusion steps, and $s$ is a small hyperparameter. Here, $\boldsymbol{m}$ is the marginal distribution of bond types in the training dataset and $\boldsymbol{m}^\top$ is the transpose of $\boldsymbol{m}$. This choice of noise schedule ensures that each bond in $\boldsymbol{E}_T$ is converged to the prior noisy distribution (i.e., the marginal distribution of bond types $\boldsymbol{m}$).

To predict the original bond matrix $\boldsymbol{E}$ from the noisy graph $\mathcal{G}^t = (\boldsymbol{A}, \boldsymbol{E}_t)$, we train a neural network $\phi_\theta\left(\mathcal{G}^t, \boldsymbol{z}_{\mathcal{G}}, \boldsymbol{z}_{\mathcal{S}}\right)$, where $\boldsymbol{z}_{\mathcal{G}}$ is extra features derived from $\mathcal{G}^t$ in DiGress and $\boldsymbol{z}_{\mathcal{S}}$ is the NMR spectra embedding from NMR-ConvFormer. We use the same graph transformer architecture as in DiGress for $\phi_\theta$ [66]. The complete training algorithm is shown in Algorithm 4.

**Sampling Procedure.** We extend DiGress [66] by conditioning on atom types $\boldsymbol{A}$ and spectra embeddings $\boldsymbol{z}_{\mathcal{S}}$. Each bond in $\boldsymbol{E}^T$ is independently sampled from the marginal distribution $\boldsymbol{m}$ to form the noisy graph $\mathcal{G}^T = (\boldsymbol{A}, \boldsymbol{E}^T)$. For each timestep $t$, we compute extra features $\boldsymbol{z}_{\mathcal{G}} = f(\mathcal{G}^t, t)$, predict bond probabilities $\hat{p}^E = \phi_\theta(\mathcal{G}^t, \boldsymbol{z}_{\mathcal{G}}, \boldsymbol{z}_{\mathcal{S}})$, and derive the posterior for each bond $e_{ij}$:

$$p_\theta(e_{ij}^{t-1} \mid \mathcal{G}^t) = \sum_e q(e_{ij}^{t-1} \mid e_{ij} = e, e_{ij}^t)\,\hat{p}_{ij}^E(e), \tag{16}$$

where $e$ can be chosed from $d_{\text{bond}}$ bond types. Then each bond in $\mathcal{G}^{t-1}$ is independently sampled according to this posterior. After $T$ steps, the denoised molecular graph $\mathcal{G}^0$ is generated. The complete algorithm is given in Algorithm 5.

# C Details of Dataset Curation

In this section, we provide details on the dataset curation process, including the data structure, preprocessing pipeline, and a summary of the statistics for each dataset.

## C.1 Dataset Structure and Preprocessing

Each data entry is represented as a tuple *(SMILES, $^1$H NMR spectrum, $^{13}$C NMR spectrum, atom features)*. The SMILES string is a sequence of characters representing a molecule [71]. Each SMILES string is canonicalized, with stereochemistry such as chiral centers and double bond configurations removed. Only molecules containing the elements C, H, O, N, S, P, F, Cl, Br, and I are retained. Duplicate entries are removed to ensure one unique SMILES per molecule.

The NMR spectra are stored as vectors, and details of the preprocessing steps are provided in Appendix C.2. Atom features include atom types $A$ and 3 ground-truth conformers $X$, which are generated using RDKit's `ETKDGv3` algorithm [1] from the SMILES string. To validate the generated 3D conformations, SMILES strings are reconstructed from the atom types and 3D coordinates using RDKit's `DetermineBonds` function [1]. Molecules that fail reconstruction are discarded. Explicit hydrogens are included to ensure accurate SMILES reconstruction, as required by `DetermineBonds`.

## C.2 NMR Spectrum Preprocessing

**Synthetic Spectra Simulation.** Synthetic spectra are generated from SMILES using MestreNova [44], with deuterated chloroform ($CDCl_3$) as the solvent. Default simulation settings are applied: $^1$H spectra ($-2$ ppm to 12 ppm, 32k points, 500.12 Hz frequency, 0.75 Hz line width) and $^{13}$C spectra ($-20$ ppm to 230 ppm, 128k points, 125.03 MHz frequency, 1.5 Hz line width, proton decoupled).

**Experimental Spectra Collection.** `SpecTeach` [65] experimental raw spectra are in `.mnova` file format, with default NMR processing steps preserved, including group delay correction, apodization in the time domain, and phase and baseline corrections in the frequency domain if exist. `NMRShiftDB2` [34] has $^{13}$C NMR spectra chemical shift lists.

**Spectra Preprocessing.** To standardize spectra from different datasets, which vary in chemical shift ranges, resolutions, and intensity scales, we adopt the formats in [4, 22] and the preprocessing method in [22]. $^1$H NMR spectra are linearly interpolated to 10,000 points in the range $[-2, 10]$ ppm, and $^{13}$C NMR spectra are interpolated to 10,000 points in the range $[-20, 230]$ ppm. Spectra outside these ranges are truncated, while shorter spectra are zero-padded. Intensities are normalized by dividing by the maximum intensity. For `SpectraBase` dataset [22] and experimental datasets, $^{13}$C NMR spectra are preprocessed into 80 binary vectors spanning $(3.42, 231.3)$ ppm.

To ensure compatibility with baseline models (i.e., Hu et al. [22] and NMR-DiGress), we also preprocess $^1$H NMR spectra into 28,000 points within the range $[-2, 12]$ ppm and $^{13}$C NMR spectra into 80 binary vectors spanning $(3.42, 231.3)$ ppm where applicable. Appendix Table 4 provides detailed preprocessing formats for each model and dataset.

Table 4: Standardized formats for preprocessed NMR spectra, specifying spectrum dimensions, chemical shift ranges, and intensity ranges.

| NMR Spectrum | Dimension | Chemical Shift (ppm) | Intensity |
|---|---|---|---|
| **CHEFNMR** | | | |
| $^1$H (Default) | 10,000 | $[-2, 10]$ | $\leq 1$ |
| $^{13}$C (`USPTO`, `SpectraNP`) | 10,000 | $[-20, 230]$ | $\leq 1$ |
| $^{13}$C (`SpectraBase`, `SpecTeach`, `NMRShiftDB2`) | 80 | $(3.42, 231.3)$ | $\{0, 1\}$ |
| **Hu et al.** [22] | | | |
| $^1$H (Default) | 28,000 | $[-2, 12]$ | $\leq 1$ |
| $^{13}$C (Default) | 80 | $(3.42, 231.3)$ | $\{0, 1\}$ |
| $^1$H (`USPTO`) | 10,000 | $[-2, 10]$ | $\leq 1$ |
| **NMR-DiGress** | | | |
| $^1$H (Default) | 10,000 | $[-2, 10]$ | $\leq 1$ |
| $^{13}$C (Default) | 80 | $(3.42, 231.3)$ | $\{0, 1\}$ |

Table 5: Summary of dataset statistics with the number of data points, heavy atoms (excluding hydrogens), atoms (including hydrogens), atom types, and solvent types reported.

| Dataset | # Data | # Heavy Atoms | # Atoms | Atom Type | Solvent |
|---|---|---|---|---|---|
| **Synthetic Datasets** | | | | | |
| SpectraBase | 141,489 | [2, 19] | [3, 59] | 4 (C, H, O, N) | $CDCl_3$ |
| USPTO | 744,602 | [5, 35] | [8, 101] | 10 (C, H, O, N, S, P, F, Cl, Br, I) | $CDCl_3$ |
| SpectraNP | 111,181 | [3, 130] | [4, 274] | 10 (C, H, O, N, S, P, F, Cl, Br, I) | $CDCl_3$ |
| **Experimental Datasets** | | | | | |
| SpecTeach | 238 | [2, 29] | [5, 59] | 7 (C, H, O, N, S, Cl, Br) | $CDCl_3$, DMSO |
| NMRShiftDB2 | 23,457 | [3, 35] | [3, 91] | 10 (C, H, O, N, S, P, F, Cl, Br, I) | >7 solvents |

## C.3 Dataset Statistics

This section provides detailed preprocessing steps and dataset statistics (Appendix Table 5).

**Synthetic Datasets.** We evaluate our models on two public benchmarks, SpectraBase [22] and USPTO [4], and our self-curated SpectraNP dataset.

SpectraBase [22] contains molecules with elements C, H, O, and N. The original dataset comprises 142,894 tuples of (Canonical nonstereo SMILES, 28,000-dimensional $^1$H NMR spectrum, 80-bin $^{13}$C NMR spectrum) along with non-overlapping split indices in a ratio of 0.8:0.1:0.1 for training, validation, and test sets. Each molecule in the dataset is unique. We remove 219 molecules with invalid $^1$H NMR spectra. After generating 3D conformations for all molecules, 141,489 valid conformations remain. The original dataset is available at https://zenodo.org/records/13892026 under the CC-BY 4.0 license.

USPTO [4] includes molecules derived from chemical reactions [41], containing elements C, H, O, N, S, P, F, Cl, Br, and I. The original dataset contains 794,403 tuples of (Canonical stereo SMILES, 10,000-dimensional $^1$H NMR spectrum, 10,000-dimensional $^{13}$C NMR spectrum). We generate 3D conformations for each molecule. The final dataset contains 744,602 entries. The original split indices are preserved, resulting in a post-filtering split ratio of 0.86:0.04:0.1 for training, validation, and test sets. The original dataset is available at https://zenodo.org/records/11611178 under the Community Data License Agreement-Sharing 1.0 (CDLA-Sharing-1.0).

SpectraNP contains 111,181 unique natural products with elements C, H, O, N, S, P, F, Cl, Br, and I. Around 31k molecules are sourced from the NPAtlas database [52], which includes small molecules from bacteria and fungi. The remaining molecules are sourced from the NP-MRD database [72], which includes natural products such as vitamins, minerals, probiotics, and small molecules from various natural sources. The dataset is randomly split into training, validation, and test sets in a ratio of 0.8:0.1:0.1.

**Experimental Datasets.** To evaluate the ability of models trained on synthetic data to generalize to experimental data, we curate two experimental datasets.

SpecTeach includes the van Bramer dataset [65]. The original dataset contains 247 tuples, but 5 compounds lack corresponding SMILES from the CAS ID, and 3 compounds have invalid experimental spectra. The final dataset comprises 238 unique tuples. Most compounds are in $CDCl_3$ solvent, with a few in DMSO solvent. The original dataset is available at https://drive.google.com/drive/folders/1R23KGk3bp6ukGCRb4U-CRuxnL6PYYBYc under CC-BY 4.0.

NMRShiftDB2 [34] is a larger-scale dataset of $^{13}$C NMR spectra in various solvents. We use a subset of 23,457 molecules excluding SMILES in the training set of USPTO. The original dataset is under the nmrshiftdb2 Database License (https://nmrshiftdb.nmr.uni-koeln.de/nmrshiftdbhtml/nmrshiftdb2datalicense.txt).

# D  Experimental Details

In this section, we provide experimental details for baseline models and CHEFNMR, including hyperparameters, training, and evaluation settings.

### D.1 Baseline Settings

We compare CHEFNMR with two existing chemical language models and introduce a graph-based model to evaluate different molecular representations for the structure elucidation task.

**Hu et al.** [22] use 28,000-dimensional $^1$H NMR spectra and 80-bin $^{13}$C NMR spectra for all datasets, except for the USPTO dataset, where 10,000-dimensional $^1$H NMR spectra are used (see Appendix Table 4). This chemical language model employs a two-stage multitask transformer to predict SMILES strings from raw 1D NMR spectra. The method pre-defines 957 substructures and pre-trains a substructure-to-SMILES transformer model on 3.1M molecules. This pre-trained model is then used to initialize a multitask transformer, which is fine-tuned on 143k data from `SpectraBase`.

For our experiments, we retrain the substructure-to-SMILES model on the same 3.1M dataset for 500 epochs. Then, we fine-tune the model on each synthetic benchmark dataset for 1500 epochs until convergence. During fine-tuning, the multitask model is initialized with the substructure-to-SMILES transformer checkpoint that achieved the lowest validation loss during the pre-training phase. Model performance is evaluated on each dataset over three independent runs, using the checkpoint with the lowest validation loss during training. Evaluation is conducted on 1 A100 GPU, with runtime varying between 30 minutes and 2 hours depending on the dataset. Other hyperparameters are set as default in the original model [22].

**Alberts et al.** [4] develop a transformer to predict stereo SMILES from text-based 1D NMR peak lists and chemical formulas. Due to the unavailability of inference code and differences in input (peak lists vs. raw spectra) and output (stereo vs. non-stereo SMILES), we report their published results on the original `USPTO` (794,403 data points) and `SpecTeach` datasets.

**NMR-DiGress** uses 10,000-dimensional $^1$H NMR spectra and 80-bin $^{13}$C NMR spectra for all datasets (see Appendix Table 4). This graph-based model, comprising 14.4M parameters, is trained on each dataset using 4 A100 GPUs for 48 hours. Evaluation is performed using the checkpoint with the highest top-1 matching accuracy on the validation set.

Table 6: Number of filtered data.

| Dataset | # Data |
|---|---|
| `SpectraBase` | 132,710 |
| `USPTO` | 673,257 |
| `SpectraNP` | 106,020 |

Notably, molecules containing aromatic nitrogens are excluded from training and evaluation (see Appendix Table 6). This is because NMR-DiGress only uses heavy atoms (excluding hydrogens) as graph nodes, and RDKit [1] fails to reconstruct SMILES strings from molecular graphs with aromatic nitrogens.

### D.2 CHEFNMR Settings

CHEFNMR use the preprocessed datasets described in Appendix Tables 4 and 5. Appendix Figure 6 illustrates the full architecture of the NMR-ConvFormer described in Section 3.1. Appendix Table 8 lists the hyperparameters and optimizer settings for CHEFNMR.

All models are trained in bf16-mixed precision. After training, we sample all molecules in the test set using the trained checkpoint for three independent runs per dataset. We select the checkpoint with the highest top-1 matching accuracy on the validation set. For experimental datasets, we use the checkpoint trained on the synthetic `USPTO` dataset with 10,000-dimensional $^1$H NMR spectra and 80-bin $^{13}$C NMR spectra. Appendix Table 7 summarizes $\sigma_{\text{data}}$, training epochs, and sampling time for CHEFNMR on each dataset. Here, $\sigma_{\text{data}}$ represents the standard deviation of the atom coordinates in the dataset. The classifier-free guidance (CFG) scale $\omega$ is set to 2 for `SpectraBase`, 1.5 for `USPTO` and `SpectraNP`, and 1 for `SpecTeach` and `NMRShiftDB2`.

Table 7: $\sigma_{\text{data}}$, training epochs, and sampling time per dataset for CHEFNMR. $\sigma_{\text{data}}$ is the standard deviation of the atom coordinates in the dataset. Sampling time is the estimated average time on 1 A100 or H100 GPU for three independent runs.

| Dataset | $\sigma_{\text{data}}$ | CHEFNMR-S | | CHEFNMR-L | |
|---|---|---|---|---|---|
| | | Train Epoch | Sample Time | Train Epoch | Sample Time |
| `SpectraBase` | 2.02 | 10k | 1h | 5k | 3h |
| `USPTO` | 2.67 | 10k | 8h | 3k | 17h |
| `SpectraNP` | 3.28 | 26k | 3.5h | 18k | 9h |

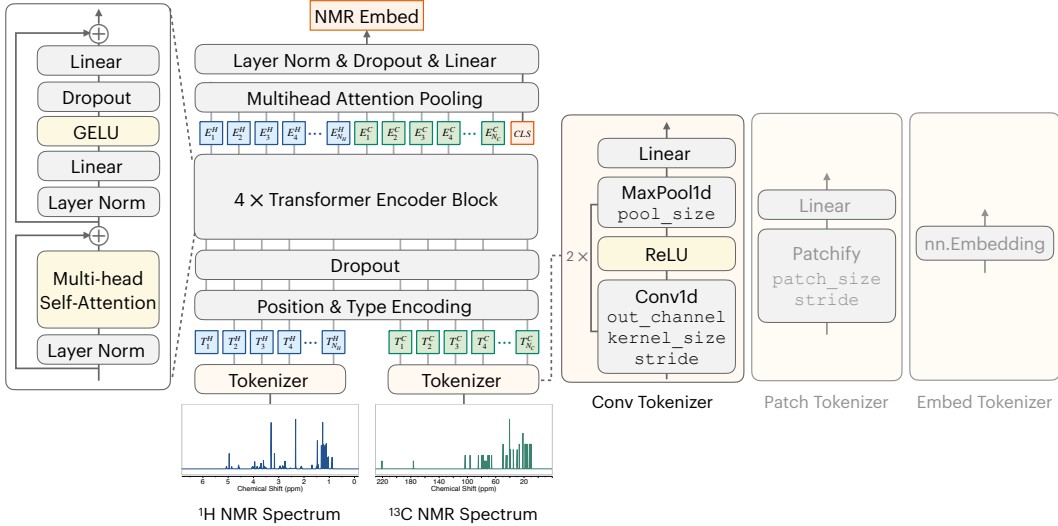

Figure 6: Details of the NMR-ConvFormer architecture. For the 10,000-dimensional $^1$H or $^{13}$C NMR spectrum, we use a convolutional tokenizer comprising two 1D convolutional layers with ReLU and max-pooling, outperforming the ViT-style patch tokenizer [67]. For the 80-bin $^{13}$C spectrum, we use learnable embeddings for each bin instead of the convolutional tokenizer. The standard transformer encoder comprises multi-head self-attention and feed-forward networks with pre-layer norm and residuals. Hyperparameters such as `out_channel` and `kernel_size` are listed in Appendix Table 8.

Table 8: CHEFNMR hyperparameters and optimizer settings.

| Parameter | CHEFNMR-S | CHEFNMR-L |
|---|---|---|
| **NMR-ConvFormer** | | |
| *General* | | |
| Encoder Dimension ($D_{\text{encoder}}$) | 256 | |
| Dropout Rate | 0.1 | |
| | | |
| *Convolutional Tokenizer* | | |
| Number of Blocks | 2 | |
| Output Channels (`out_channel`) | [64, 128] | |
| Kernel Sizes (`kernel_size`) | [5, 9] | |
| Stride Sizes (`stride`) | [1, 1] | |
| Max Pooling Sizes (`pool_size`) | [8, 12] | |
| | | |
| *Transformer Encoder* | | |
| Positional Encoding | Learnable | |
| Type Encoding | Learnable | |
| Number of Blocks | 4 | |
| Number of Attention Heads | 8 | |
| Head Dimension | 32 | |
| MLP Ratio | 4 | |
| | | |
| *Pooling* | | |
| Pooling Strategy | Multihead Attention Pooling | |
| | | |
| **DiT** | | |
| Number of Blocks | 12 | 24 |
| Number of Attention Heads | 8 | 16 |
| Hidden Dimension | 768 | 1024 |
| MLP Ratio | 4 | |
| | | |
| **Optimizer (Adam)** | | |
| Learning Rate | 1e-4 | |
| Adam $\beta_1$ | 0.9 | |
| Adam $\beta_2$ | 0.95 | |
| Adam $\epsilon$ | 1e-8 | |

Table 9: Ablation study on NMR-ConvFormer components.

| Configuration | Tokenizer | # Tokens | Transformer | Pooling | Dropout | Acc@1% ↑ | Sim@1 ↑ |
|---|---|---|---|---|---|---|---|
| Base (CHEFNMR-S) | Conv | 183 | Y | MAP | 0.1 | **69.15**±.08 | **0.807**±.002 |
| **Tokenizer Ablation** | | | | | | | |
| – Conv Tokenizer | Patch | 183 | Y | MAP | 0.1 | 61.78±.18 | 0.756±.000 |
| – Token Count Reduction | Conv | 121 | Y | MAP | 0.1 | 66.28±.18 | 0.789±.001 |
| **Transformer Ablation** | | | | | | | |
| – Transformer Block | Conv | 183 | N | MAP | 0.1 | 68.12±.17 | 0.797±.001 |
| **Pooling Ablation** | | | | | | | |
| – MAP Pooling | Conv | 183 | Y | Flatten | 0.1 | 62.97±.14 | 0.758±.002 |
| **Dropout Ablation** | | | | | | | |
| – Dropout | Conv | 183 | Y | MAP | 0.0 | 65.48±.06 | 0.777±.001 |

# E   Additional Ablation Studies

In this section, we provide additional ablation studies to evaluate the impact of the NMR-ConvFormer components and the impact of different NMR spectra on CHEFNMR's performance.

## E.1   Ablation Studies for NMR-ConvFormer

We perform extensive ablation studies to evaluate the contributions of key components in the NMR-ConvFormer on the `SpectraBase` dataset using CHEFNMR-S. The results are summarized in Table 3 of the main paper, with detailed configurations provided here.

Some of the modifications in Appendix Table 9 are: **– Conv Tokenizer**: The convolutional tokenizer is replaced with a patch tokenizer (see Appendix Figure 6) using `patch_size = 192` and `stride = 96` to maintain the same token count. **– Token Count Reduction**: The number of tokens is reduced from 183 to 121 by increasing max pooling sizes (`pool_size` in Appendix Table 8) from [8, 12] to [12, 20]. **– MAP Pooling**: The MAP pooling layer is replaced with a flattening layer, which reshapes the transformer encoder's output from $(\text{batch\_size}, T, D_{\text{encoder}})$ to $(\text{batch\_size}, T \times D_{\text{encoder}})$.

We find that within the NMR-ConvFormer, the convolutional tokenizer outperforms the patch tokenizer, likely due to its ability to capture local features more effectively. Reducing the number of tokens leads to a drop in performance. The MAP pooling layer is more effective than flattening for aggregating features. Dropout regularization is necessary to prevent overfitting.

## E.2   Ablation Studies for Different NMR Spectra

We investigate the impact of using different NMR spectra ($^1$H NMR, $^{13}$C NMR, or both) on model performance on the `SpectraBase` dataset. The results are presented in Appendix Table 10. CHEFNMR

Table 10: Performance on `SpectraBase` using different NMR spectra.

| Spectrum | Model | Top-1 Acc% ↑ | Top-1 Sim ↑ | Top-5 Acc% ↑ | Top-5 Sim ↑ | Top-10 Acc% ↑ | Top-10 Sim ↑ |
|---|---|---|---|---|---|---|---|
| $^{13}$C | Hu et al. | 4.50 ±.15 | 0.296±.000 | 12.92±.11 | 0.460±.001 | 18.27±.14 | 0.523±.001 |
| | NMR-DiGress | 10.87±.08 | 0.314±.001 | 19.35±.03 | 0.438±.001 | 23.00±.02 | 0.477±.001 |
| | CHEFNMR-S | 26.69±.06 | 0.469±.001 | 39.90±.33 | 0.612±.001 | 44.59±.07 | 0.651±.000 |
| | CHEFNMR-L | **30.28**±.15 | **0.510**±.002 | **47.33**±.14 | **0.672**±.000 | **53.53**±.30 | **0.718**±.001 |
| $^1$H | Hu et al. | 31.12±.15 | 0.569±.001 | 48.64±.42 | 0.725±.001 | 54.92±.26 | 0.771±.001 |
| | NMR-DiGress | 31.13±.13 | 0.521±.001 | 48.73±.24 | 0.682±.001 | 54.52±.18 | 0.723±.001 |
| | CHEFNMR-S | 57.97±.25 | 0.720±.002 | 72.64±.22 | 0.841±.002 | 76.58±.23 | 0.867±.002 |
| | CHEFNMR-L | **59.37**±.20 | **0.739**±.001 | **74.91**±.11 | **0.857**±.001 | **79.20**±.09 | **0.884**±.000 |
| $^{13}$C + $^1$H | Hu et al. | 45.24±.18 | 0.686±.001 | 62.37±.08 | 0.815±.001 | 67.38±.05 | 0.847±.001 |
| | NMR-DiGress | 43.56±.30 | 0.625±.002 | 62.47±.20 | 0.779±.002 | 68.39±.35 | 0.817±.001 |
| | CHEFNMR-S | 69.15±.08 | 0.807±.002 | 82.09±.24 | 0.904±.002 | 85.30±.04 | 0.922±.000 |
| | CHEFNMR-L | **72.04**±.02 | **0.833**±.000 | **85.24**±.10 | **0.923**±.001 | **88.20**±.07 | **0.940**±.000 |

consistently and significantly outperforms the baselines with $^1$H or/and $^{13}$C spectra. The combination of $^1$H and $^{13}$C spectra provides complementary information that yields the best performance.

# F    Additional Results and Analysis

This section provides additional results and analysis across datasets, demonstrating the state-of-the-art performance and generalization ability of CHEFNMR. Appendix F.1 reports the Average Minimum RMSD metrics for generated 3D structures. Appendix F.2 analyzes CHEFNMR's generalization ability to unseen molecular scaffolds and different solvents in NMR spectra. Appendix F.3 presents a systematic failure mode analysis of CHEFNMR by molecular structures and domain shift between synthetic and real spectra. Appendix F.4 and F.5 provide additional qualitative examples on synthetic and experimental datasets respectively.

## F.1    RMSD Metric

Since CHEFNMR generates atomic 3D coordinates, we additionally report the top-$k$ Average Minimum RMSD (AMR) of heavy atoms in Appendix Table 11. The RMSD for each predicted structure is computed against three ground-truth conformers and taken as the minimum value. We then select the minimum RMSD among the top-$k$ predictions for each molecule and average across all molecules to obtain the top-$k$ AMR Although RMSD is not size-independent and thus less interpretable, the obtained results are reasonable given the dataset complexity.

Table 11: Average Minimum RMSD in Å.

| Dataset | Top-1↓ | Top-5↓ | Top-10↓ |
|---|---|---|---|
| `SpectraBase` | 3.26 | 2.87 | 2.71 |
| `USPTO` | 4.59 | 4.14 | 3.96 |
| `SpectraNP` | 5.50 | 5.12 | 4.97 |
| `SpecTeach` | 2.10 | 1.71 | 1.56 |
| `NMRShiftDB2` | 3.23 | 2.83 | 2.68 |

## F.2    Generalization Analysis

In this section, we analyze CHEFNMR's generalization ability to unseen molecular scaffolds (Appendix F.2.1) and different solvents (Appendix F.2.2) in NMR spectra.

### F.2.1    Generalization to Unseen Molecular Scaffolds

We evaluate CHEFNMR's generalization ability to unseen molecular scaffolds by creating test subsets with scaffolds not present in the training sets. Appendix Table 12 shows the subsets are relatively chemically dissimilar to the training sets, based on Scaffold similarity (Scaff), fingerprint-based Tanimoto Similarity to a nearest neighbor (SNN) [51], and the absolute difference of standard deviation of

Table 12: Similarity analysis between training and test splits across datasets. Unseen: test subsets with scaffolds unseen during training. Scaff: Scaffold similarity; SNN: Tanimoto similarity to nearest neighbor. $|\sigma_{\text{train}} - \sigma_{\text{test}}|$: absolute difference of std of atom coordinates between training and test sets.

| Train Set | Test (Sub)set | #Test Set Data | Scaff↑ | SNN↑ | $|\sigma_{\text{train}} - \sigma_{\text{test}}|$↓ |
|---|---|---|---|---|---|
| `SpectraBase` | `SpectraBase` | 14137 | 0.959 | 0.659 | 0.004 |
| `SpectraBase` | `SpectraBase` (Unseen) | 3770 (26.7%) | 0.000 | 0.657 | 0.025 |
| `USPTO` | `USPTO` | 73059 | 0.980 | 0.698 | 0.005 |
| `USPTO` | `USPTO` (Unseen) | 14711 (20.1%) | 0.000 | 0.656 | 0.174 |
| `SpectraNP` | `SpectraNP` | 10952 | 0.894 | 0.722 | 0.012 |
| `SpectraNP` | `SpectraNP` (Unseen) | 2897 (26.5%) | 0.000 | 0.655 | 0.144 |

Table 13: Performance on unseen scaffold test subsets across synthetic datasets.

| Dataset | Model | Top-1 | | Top-5 | | Top-10 | |
|---|---|---|---|---|---|---|---|
| | | Acc% ↑ | Sim ↑ | Acc% ↑ | Sim ↑ | Acc% ↑ | Sim ↑ |
| SpectraBase | Hu et al. | 39.59±.26 | 0.653±.002 | 55.43±.64 | 0.776±.003 | 59.85±.59 | 0.807±.003 |
| | NMR-DiGress | 43.10±.29 | 0.607±.003 | 59.70±.22 | 0.759±.001 | 64.83±.24 | 0.793±.002 |
| | CHEFNMR-S | 64.09±.32 | 0.758±.003 | 77.35±.20 | 0.871±.001 | 80.73±.22 | 0.892±.001 |
| | CHEFNMR-L | **66.40±.31** | **0.785±.002** | **80.24±.33** | **0.891±.001** | **83.75±.24** | **0.912±.001** |
| USPTO | Hu et al. | 25.80±.22 | 0.590±.001 | 41.58±.27 | 0.738±.001 | 47.12±.27 | 0.778±.002 |
| | NMR-DiGress | 12.93±.23 | 0.388±.003 | 25.27±.22 | 0.596±.002 | 31.42±.12 | 0.653±.001 |
| | CHEFNMR-S | 66.68±.04 | 0.814±.002 | 81.31±.08 | 0.921±.000 | 84.81±.04 | **0.938±.000** |
| | CHEFNMR-L | **67.78±.25** | **0.836±.003** | **81.79±.11** | **0.925±.000** | **85.26±.13** | **0.941±.001** |
| SpectraNP | Hu et al. | 7.68±.36 | **0.478±.001** | 16.16±.27 | 0.623±.001 | 20.26±.27 | 0.659±.000 |
| | NMR-DiGress | 1.59±.10 | 0.222±.004 | 4.50±.24 | 0.389±.002 | 6.18±.19 | 0.436±.001 |
| | CHEFNMR-S | 21.40±.44 | 0.417±.001 | 35.30±.20 | 0.641±.001 | 40.14±.09 | 0.696±.000 |
| | CHEFNMR-L | **21.82±.24** | **0.477±.002** | **37.04±.66** | **0.694±.001** | **42.69±.31** | **0.741±.002** |

atom coordinates between training and test sets. Appendix Table 13 shows the performance on these unseen scaffold subsets with different models. CHEFNMR still significantly outperforms baselines across these subsets, demonstrating its generalization ability.

### F.2.2 Generalization Across Solvents in Experimental Spectra

We report top-10 zero-shot accuracy on 2k experimental 13C spectra paired with solvent information from `NMRShiftDB2` [34] in Appendix Table 14. CHEFNMR trained on synthetic 13C spectra with $CDCl_3$ solvent shows generalization ability to various solvents except for $C_5D_5N$ and $CD_3CN$.

Table 14: Top-10 zero-shot accuracy of CHEFNMR on `NMRShiftDB2` [34] across different solvents.

| Solvent | #Molecules | Top-10 Zero-shot Accuracy |
|---|---|---|
| $CDCl_3$ | 1498 | 0.263 |
| DMSO | 232 | 0.323 |
| $CD_3OD$ | 197 | 0.102 |
| $C_5D_5N$ | 57 | 0.035 |
| $C_6D_6$ | 48 | 0.188 |
| $D_2O$ | 36 | 0.500 |
| $CCl_4$ | 33 | 0.788 |
| $CD_3CN$ | 11 | 0.000 |
| $(CD_3)_2CO$ | 2 | 0.500 |

### F.3 Failure Mode Analysis

In this section, we systematically analyze the failure modes of CHEFNMR by molecular structures (Appendix F.3.1) and domain shift between synthetic and real spectra (Appendix F.3.2).

### F.3.1 Failure Mode by Molecular Structures

Failure rate is defined as the proportion of molecules where the model fails to generate the correct structure within the top-10 predictions. We analyze the failure rate by molecular structures, including the number of atoms, number of rings, largest ring size, and functional groups of our CHEFNMR on the synthetic `SpectraNP` (Appendix Table 15 and 16), and experimental `SpecTeach` (Appendix Table 17 and 18) and `NMRShiftDB2` (Appendix Table 19 and 20) datasets.

On all datasets, the model fails more often on molecules with the most atoms or the largest number of rings due to less training data and increasing complexity of spectra and structures for larger molecules. However, the model is not systematically significantly failing in a specific functional group category.

Table 15: Failure rate analysis on synthetic `SpectraNP` dataset by molecular properties.

| **By Total Atoms (including hydrogens)** | | | | | | |
|---|---|---|---|---|---|---|
| Total Atoms | ≤40 | 41–80 | 81–120 | 121–160 | 161–200 | >200 |
| Total Molecules | 2265 | 6168 | 1818 | 539 | 140 | 22 |
| Failure Rate | 0.296 | 0.315 | 0.411 | 0.492 | 0.707 | 1.000 |
| **By Heavy Atoms** | | | | | | |
| Heavy Atoms | ≤20 | 21–40 | 41–60 | 61–80 | 81–100 | >100 |
| Total Molecules | 2099 | 6563 | 1750 | 415 | 110 | 15 |
| Failure Rate | 0.298 | 0.311 | 0.431 | 0.542 | 0.773 | 1.000 |
| **By Number of Rings** | | | | | | |
| Ring Count | 0 | 1–2 | 3–4 | 5–6 | 7–8 | >8 |
| Total Molecules | 559 | 3070 | 4437 | 2102 | 560 | 224 |
| Failure Rate | 0.250 | 0.317 | 0.332 | 0.381 | 0.407 | 0.571 |
| **By Largest Ring Size** | | | | | | |
| Largest Ring | No rings | 3–5 | 6 | 7–8 | >8 | |
| Total Molecules | 559 | 483 | 8061 | 778 | 1071 | |
| Failure Rate | 0.250 | 0.383 | 0.322 | 0.382 | 0.490 | |

Table 16: Failure rate by functional groups on synthetic `SpectraNP` dataset.

| Category | Carbonyls | Amides | Esters | Ethers | Alcohols | Halogen | Sulfur | Nitrogen |
|---|---|---|---|---|---|---|---|---|
| Total Molecules | 8148 | 1479 | 4389 | 8048 | 6861 | 477 | 138 | 3046 |
| Failure Rate | 0.352 | 0.411 | 0.354 | 0.342 | 0.355 | 0.356 | 0.341 | 0.386 |

Table 17: Failure rate analysis on experimental `SpecTeach` dataset by molecular properties.

| **By Total Atoms (including hydrogens)** | | | | | | |
|---|---|---|---|---|---|---|
| Total Atoms | ≤10 | 11–15 | 16–20 | 21–25 | 26–30 | >30 |
| Total Molecules | 10 | 60 | 76 | 48 | 26 | 18 |
| Failure Rate | 0.000 | 0.217 | 0.289 | 0.271 | 0.500 | 0.556 |
| **By Heavy Atoms** | | | | | | |
| Heavy Atoms | ≤5 | 6–10 | 11–15 | 16–20 | >25 | |
| Total Molecules | 38 | 154 | 37 | 8 | 1 | |
| Failure Rate | 0.105 | 0.240 | 0.676 | 0.500 | 1.000 | |
| **By Number of Rings** | | | | | | |
| Ring Count | 0 | 1 | >1 | | | |
| Total Molecules | 157 | 69 | 12 | | | |
| Failure Rate | 0.210 | 0.449 | 0.583 | | | |
| **By Largest Ring Size** | | | | | | |
| Largest Ring | No rings | 3–5 | 6 | | | |
| Total Molecules | 157 | 4 | 77 | | | |
| Failure Rate | 0.210 | 0.250 | 0.481 | | | |

Table 18: Failure rate by functional groups on experimental `SpecTeach` dataset.

| Category | Carbonyls | Amides | Esters | Ethers | Alcohols | Halogen | Nitrogen |
|---|---|---|---|---|---|---|---|
| Total Molecules | 125 | 3 | 46 | 58 | 43 | 24 | 24 |
| Failure Rate | 0.304 | 0.667 | 0.435 | 0.448 | 0.256 | 0.208 | 0.542 |

Table 19: Failure rate analysis on experimental `NMRShiftDB2` dataset by molecular properties.

| By Total Atoms (including hydrogens) | | | | | |
|---|---|---|---|---|---|
| Total Atoms | $\leq 20$ | 21–40 | 41–60 | 61–80 | >80 |
| Total Molecules | 6484 | 13685 | 2806 | 417 | 65 |
| Failure Rate | 0.351 | 0.689 | 0.916 | 0.947 | 0.985 |
| By Heavy Atoms | | | | | |
| Heavy Atoms | $\leq 10$ | 11–20 | 21–30 | >30 | |
| Total Molecules | 5591 | 13855 | 3504 | 507 | |
| Failure Rate | 0.326 | 0.664 | 0.917 | 0.982 | |
| By Number of Rings | | | | | |
| Ring Count | 0 | 1–2 | 3–4 | 5–6 | 7–8 |
| Total Molecules | 2955 | 14763 | 5192 | 514 | 33 |
| Failure Rate | 0.334 | 0.592 | 0.862 | 0.977 | 1.000 |
| By Largest Ring Size | | | | | |
| Largest Ring | No rings | 3–5 | 6 | 7–8 | >8 |
| Total Molecules | 2955 | 3031 | 16552 | 682 | 237 |
| Failure Rate | 0.334 | 0.630 | 0.666 | 0.871 | 0.916 |

Table 20: Failure rate by functional groups on experimental `NMRShiftDB2` dataset.

| Category | Carbonyls | Amides | Esters | Ethers | Alcohols | Halogen | Sulfur | Nitrogen |
|---|---|---|---|---|---|---|---|---|
| Total Molecules | 11096 | 2562 | 4320 | 9699 | 3756 | 6248 | 1207 | 12074 |
| Failure Rate | 0.681 | 0.713 | 0.726 | 0.723 | 0.759 | 0.528 | 0.725 | 0.674 |

### F.3.2 Domain Shift between Synthetic and Real Spectra

To quantify the domain shift between synthetic and real spectra, we simulate synthetic spectra for molecules in the `SpecTeach` dataset using MestReNova, and compute the cosine similarity between synthetic and real spectra following [4]. We also report the average cosine similarity of successful and failed predictions on the `SpecTeach`. Appendix Table 21 shows that 10,000-dimensional $^1$H NMR spectra have significantly lower cosine similarity than 80-bin 13C NMR spectra, indicating a need for more robust representation of $^1$H NMR spectra. In addition, failed predictions have lower similarity between synthetic and real $^1$H spectra. We note that it is non-trivial to develop systematic metrics to quantify the spectra domain shift, and we leave it to future work.

Table 21: Cosine similarity between synthetic and experimental spectra of molecules in `SpecTeach`, and successful and failed predictions by CHEFNMR.

| Type | Count | Cos Sim (1H) | Cos Sim (13C) |
|---|---|---|---|
| All | 238 | 0.174±.195 | 0.743±.214 |
| Success | 167 | 0.197±.199 | 0.747±.206 |
| Fail | 71 | 0.119±177 | 0.735±.234 |

### F.4 Additional Qualitative Results on Synthetic Spectra

We present more examples of CHEFNMR's predictions on the synthetic `SpectraBase` dataset (Appendix Figure 7), the synthetic `USPTO` dataset (Appendix Figure 8), and the synthetic `SpectraNP` dataset (Appendix Figure 9). The quantitative results demonstrate that CHEFNMR effectively elucidates diverse chemical structures across various synthetic datasets.

### F.5 Additional Qualitative Results on Experimental Spectra

We present additional examples of CHEFNMR's zero-shot predictions on experimental datasets, including the `SpecTeach` dataset (Appendix Figure 10) and the `NMRShiftDB2` dataset (Appendix Figure 11). These results highlight CHEFNMR's robustness to experimental variability, such as differences in solvents, impurities, and baseline noise.

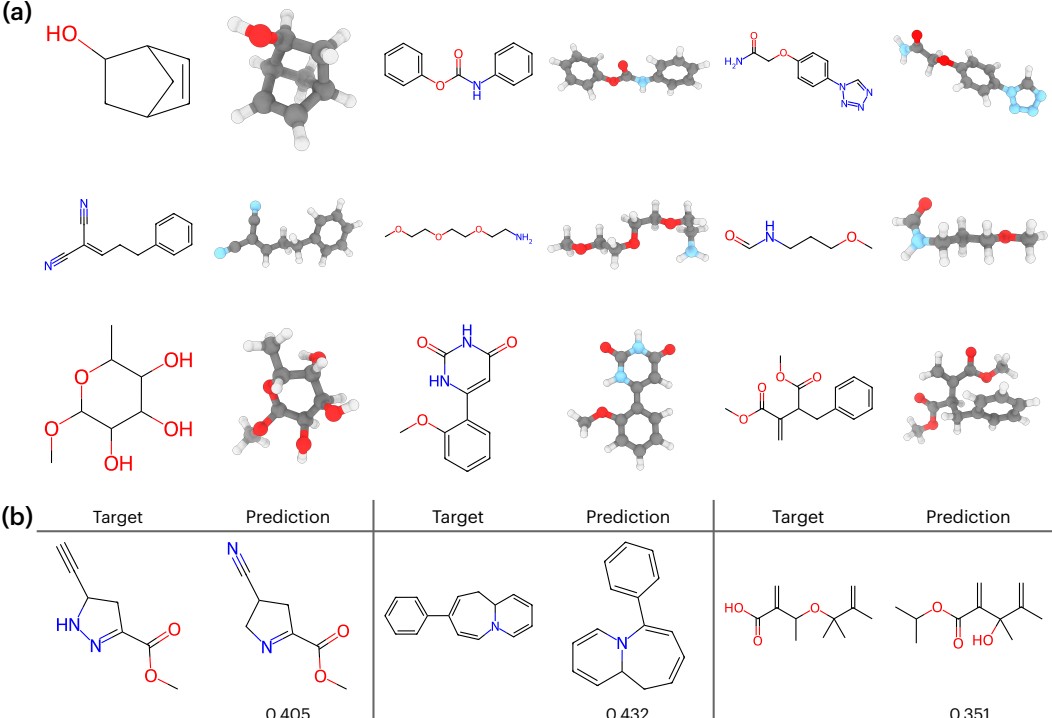

Figure 7: Examples of CHEFNMR's predictions on the synthetic `SpectraBase` dataset. **(a)** Correctly predicted structures in top-1 predictions. **(b)** Incorrect top-1 predictions with corresponding Tanimoto similarity scores.

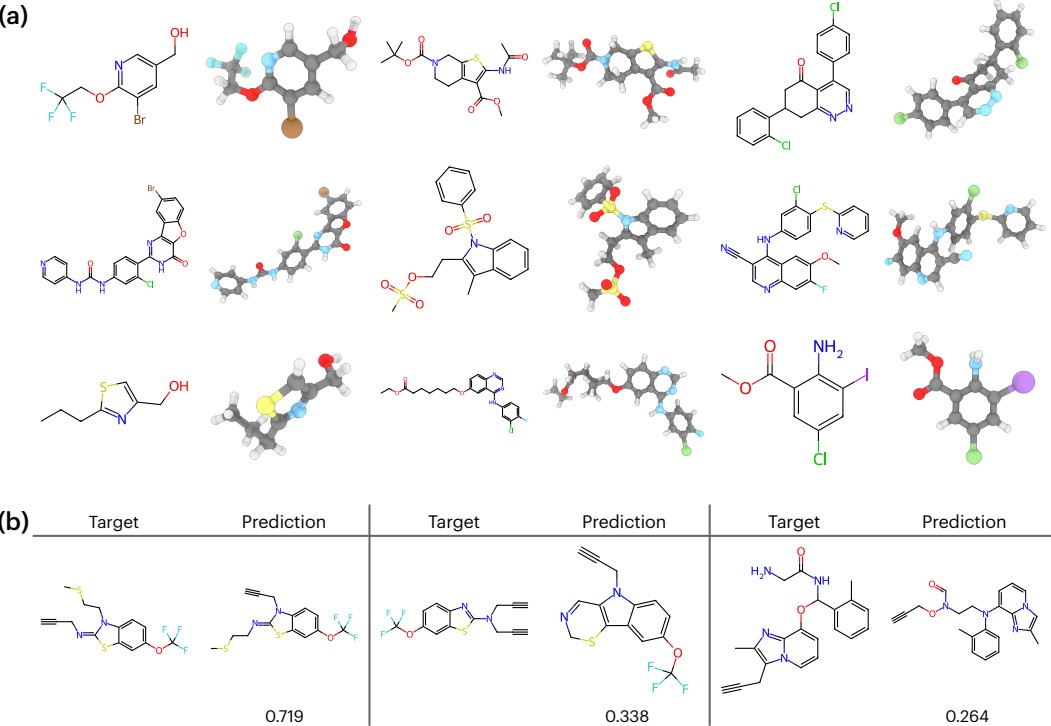

Figure 8: Examples of CHEFNMR's predictions on the synthetic `USPTO` dataset. **(a)** Correctly predicted structures in top-1 predictions. **(b)** Incorrect top-1 predictions with corresponding Tanimoto similarity scores.

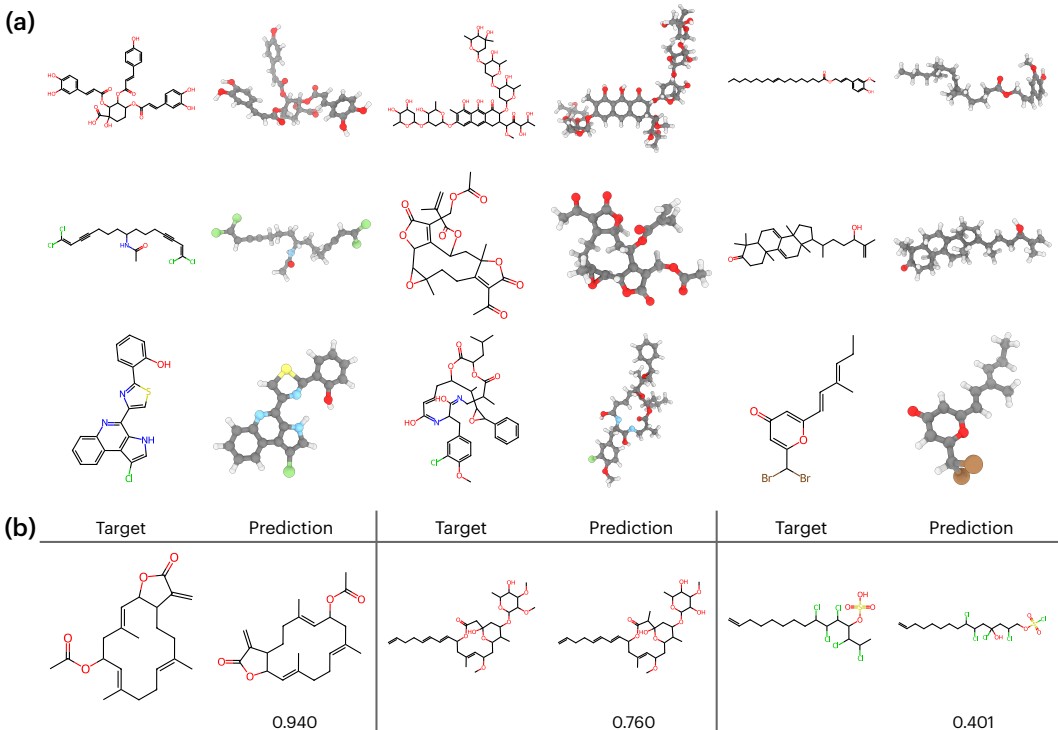

Figure 9: Additional examples of CHEFNMR's predictions on the synthetic `SpectraNP` dataset. **(a)** Correctly predicted structures in top-1 predictions. **(b)** Incorrect top-1 predictions with corresponding Tanimoto similarity scores.

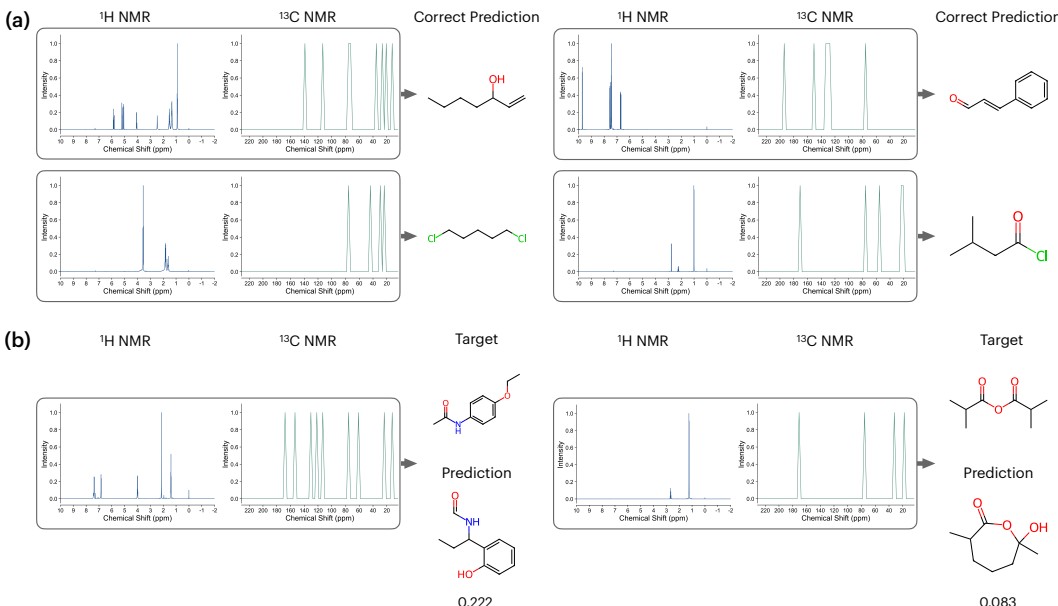

Figure 10: Examples of experimental spectra from the `SpecTeach` dataset and corresponding CHEFNMR predictions. **(a)** Correct top-1 predictions. **(b)** Incorrect top-1 predictions with corresponding Tanimoto similarity scores.

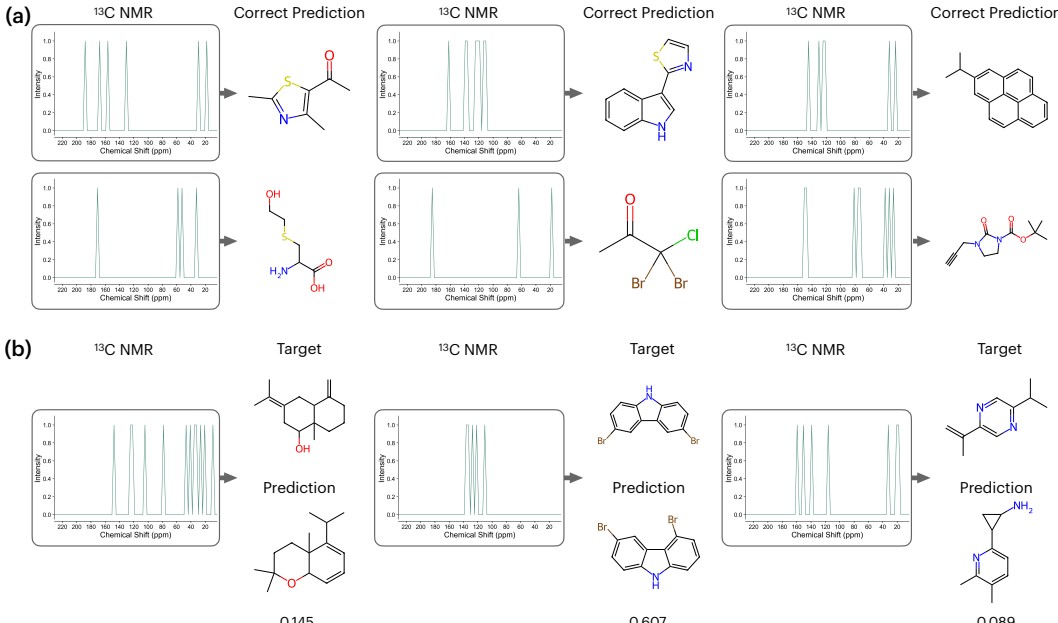

Figure 11: Examples of experimental spectra from the `NMRShiftDB2` dataset and corresponding CHEFNMR predictions. **(a)** Correct top-1 predictions. **(b)** Incorrect top-1 predictions with corresponding Tanimoto similarity scores.

