# OpenReview forum: "Atomic Diffusion Models for Small Molecule Structure Elucidation from NMR Spectra"
_NeurIPS.cc/2025/Conference — NeurIPS 2025 poster_

### Official Review · Reviewer_BFWB · 2025-06-11

**Clarity:** 4
**Significance:** 2
**Originality:** 3
**Rating:** 5
**Confidence:** 3

**Summary:**

NMRDiff3D is proposed to predict the 3D structure of molecules from 1D NMR spectra. The model uses a transformer encoder over a 1D convolutional layer to predict 3D atom coordinates. Diffusion is used as the training strategy, and data augmentation is applied to learn SE(3) equivariance by scaling up. Results show that NMRDiff3D outperforms previous methods on synthetic and experimental datasets.

**Questions:**

I have questions linked to my concerns above. I would be happy to increase the grades if these questions are answered.


1. Can you comment on the accuracy of the synthetic data generation? How reliable is the used MestreNova software for natural products? Is there a way to verify the generated structures?
2. Why did you remove stereochemistry from the dataset? How does it affect the evaluation of the model?
3. Can you report the 2D and 3D structural similarity between the train and test splits? If these splits are similar, can you run experiments on distant splits to reveal the generalization of the model and the benchmarks?
4. How long does it take to train one NMRDiff3D model?

**Ethical Concerns:**

["NO or VERY MINOR ethics concerns only"]

**Final Justification:**

The use of diffusion and the designed architectures is novel in the task. The contribution of each component is well-studied in ablation studies. The experiments support the claims made in the paper, and I believe this can be a high-impact paper in NMR field.

My concerns about the applicability domain are addressed by the authors in the rebuttals experiments.

I updated my rating accordingly.

**Limitations:**

Yes

**Quality:**

3

**Strengths And Weaknesses:**

I congratulate the authors. I truly enjoyed reading the work.

Originality:
- First approach that uses diffusion to predict 3D coordinates from 1D NMR spectra. Diffusion is a well-motivated choice for this task due to its recent success in 3D generation.
- The developed deep learning architecture is a well-motivated architecture and fits the data modality.
- Learning equivariance at scale, despite not being a novel idea, is a good choice to simplify the model.


Clarity:
- Very well-structured and written paper. It was easy to follow top to bottom, although I am not an expert in NMR spectroscopy.

Quality:
- Experimental pipelines are well thought out and executed.


Significance
- The performance gain over the benchmark is significant.
- It can replace the existing methods for 3D structure prediction from 1D NMR spectra. I cannot estimate the size of this potential, though, due to my background.


**Weaknesses**

I believe the work is already in good shape. I have several concerns about the applicability domain and evaluation of the model. I will happily update my grades if these concerns are addressed.

Quality

1. (L260) The model size indeed increases the performance, but a big performance boost is already observed with NMRDiff3D-M. This can be emphasized. I believe this would help readers appreciate the quality of the model and the ideas behind it better. Authors can also change the naming to NMRDiff3D-S vs NMRDiff3D-L, as "M" naming hints at the existence of an "S" model, which is not the case.
2. The authors generate the synthetic data via MestreNova. However, the applicability of this method to generate 3D structures of natural products is not discussed.
3. Stereochemistry is removed when generating the dataset. However, the studied task is in 3D by nature, and natural products are *known* to have rich stereochemistry. This can make the evaluation task artificially easier.

Significance
1. The structural and 3D similarity between the train and test splits is not reported. This is very important to measure the generalization of the model, as well as its real-world applicability.

Clarity

1. (L279) The ablation study discussion can be revisited to add more nuance and emphasize the magnitude of the effects. For instance, not all ConvFormer components are equally important (some cause only a 1% drop in accuracy), and this can be highlighted. The comparison to the patch tokenizer can be moved to the main text from the appendix.
2. Not a weakness, but a suggestion: authors can list their key contributions at the end of the intro, as this is typical in NeurIPS papers and helps the reader to follow the paper.

---

> ### Author Rebuttal · Authors · 2025-07-31
>
> Thank you for your thoughtful and extensive feedback. We are especially grateful for your recognition of the technical novelty, performance, and clarity of our paper. Below, we address your clarifications and concerns.
>
> **[Accuracy of synthetic data generation]** MestreNova’s NMR prediction is a proprietary algorithm that uses an ensemble approach combining machine learning models with rule-based methods. While the exact implementation details and quantitative accuracy benchmarks are not publicly available, MestreNova is widely used in both academic and industrial settings and is considered a gold standard for NMR spectral prediction and analysis. Given the lack of large-scale experimental NMR datasets, we believe MestreNova provides sufficiently accurate training data for our task.
>
> **[Stereochemistry]** We agree that including stereochemistry significantly increases the complexity of the structure elucidation problem, and we will highlight this complexity in our revisions. We initially did not include stereochemical information as 2D NMR spectra, degradation analysis, Marfey's analysis, or Mosher ester analysis are usually required to determine absolute configuration at chiral centers. To investigate the impact of stereochemistry, also suggested by reviewer 1WdQ, we train and evaluate NMRDiff3D-M on the USPTO, including stereochemistry as Alberts et al. [1]. Our model still consistently outperforms Alberts et al. with single 1H spectra or 1H and 13C spectra with sterechemistry information. We note that including stereochemistry indeed degrades the top-1 accuracy, but interestingly, our model can mostly predict the correct stereoisomer within the top-5 predictions.
>
> |Spectra|Model| Top-1| Top-5| Top-10|
> |-|-|-|-|-|
> ||| Acc% ↑ / Acc% (w/o stereo) ↑ | Acc% ↑ / Acc% (w/o stereo) ↑ | Acc% ↑ / Acc% (w/o stereo) ↑ |
> | 1H | Alberts et al. | 64.99 / NA | 81.94 / NA | 84.07 / NA |
> | 1H | NMRDiff3D-M | **76.88 / 84.71** | **89.49 / 92.27** | **92.43 / 93.92** |
> | 1H + 13C | Alberts et al. | 73.38 / NA | 87.94 / NA |  89.98 / NA |
> | 1H + 13C | NMRDiff3D-M | **78.08 / 86.18** | **90.27 / 93.05** | **93.05 / 94.54** |
>
>
> **[Generalization to chemically dissimilar structures]** Thanks for your suggestion on providing similarity analysis between the train and test splits, as also suggested by reviewer 1WdQ. We compute the Scaffold similarity (Scaff) and fingerprint-based Tanimoto Similarity to a nearest neighbor (SNN) metrics [2] between the test set and training set. We denote $\sigma_\textrm{dataset}$ as the standard deviation of the atom coordinates in the dataset, and compute the absolute difference of std between the test set and training set, as shown in the table below. All metrics indicate that the training and test splits of all datasets are relatively similar.
>
> | Train set | Test (Sub)set | # Test set data | Scaff ↑| SNN ↑| $\|\sigma_\textrm{train} - \sigma_\textrm{test}\|$ ↓|
> |-|-|-|-|-|-|
> | SpectraBase | SpectraBase | 14178| 0.959 | 0.659 | 0.004 |
> | SpectraBase | SpectraBase (Unseen Scaffolds) | 3786 (26.7%) | 0.000 | 0.657 | 0.025 |
> | USPTO | USPTO | 73438 | 0.980 | 0.698 | 0.005 |
> | USPTO | USPTO (Unseen Scaffolds) | 14796 (20.1%) | 0.000 | 0.656 |0.174|
> | SpectraNP | SpectraNP | 10925 | 0.894 | 0.722 | 0.012 |
> | SpectraNP | SpectraNP (Unseen Scaffolds) | 2898 (26.5%) | 0.000 | 0.655 | 0.144|
>
> We then evaluate all models on test subsets with scaffolds unseen during training, which are relatively chemically dissimilar to the training sets according to the metrics in the table above. NMRDiff3D still significantly outperforms baselines across these subsets, demonstrating its generalization ability.
>
> |Dataset| Model| Top-1| Top-5| Top-10|
> |-|-|-|-|-|
> ||| Acc% ↑ / Sim ↑ | Acc% ↑ / Sim ↑ | Acc% ↑ / Sim ↑ |
> | SpectraBase | Hu et al. | 39.59 / 0.653  | 55.43 / 0.776 | 59.85 / 0.807 |
> | SpectraBase | NMR-DiGress | 43.10 / 0.607 | 59.70 / 0.759 | 64.83 / 0.793 |
> | SpectraBase | NMRDiff3D-M | 65.06 / 0.746 | 75.22 / 0.846 | 78.01 / 0.868 |
> | SpectraBase | NMRDiff3D-L | **68.80 / 0.789** | **77.59 / 0.867** | **80.35 / 0.886** |
> | USPTO | Hu et al. | 15.86 / 0.503 | 23.95 / 0.626 | 26.73 / 0.657 |
> | USPTO | NMR-DiGress | 12.93 / 0.388 | 25.27 / 0.596 | 31.42 / 0.653 |
> | USPTO | NMRDiff3D-M | 74.52 / 0.834 | 84.25 / 0.916 | 86.85 / 0.932 |
> | USPTO | NMRDiff3D-L | **75.75 / 0.844** | **84.71 / 0.918** | **87.05 / 0.933** |
> | SpectraNP | Hu et al.| 6.29 / 0.450 | 13.56 / 0.594 | 17.35 / 0.633 |
> | SpectraNP | NMR-DiGress | 1.59 / 0.222 | 4.50 / 0.389 | 6.18 / 0.436 |
> | SpectraNP | NMRDiff3D-M | 30.78 / 0.473 | 42.27 / 0.645 | 45.66 / 0.686 |
> | SpectraNP | NMRDiff3D-L | **33.30 / 0.504** | **43.04 / 0.655** | **46.31 / 0.692** |
>
>
> **[Minor questions or suggestions]**
>
> > (L260) you should emphasize the big performance boost is already observed with NMRDiff3D-M. Authors can also change the naming to NMRDiff3D-S vs NMRDiff3D-L, as "M" naming hints at the existence of an "S" model, which is not the case.
>
> Thank you for pointing this out. We agree that NMRDiff3D-M has already achieved a strong performance, and we will emphasize this in the revision to highlight the strength of the core design. We also appreciate the naming suggestion and will rename NMRDiff3D-M to NMRDiff3D-S for clarity.
> > (L279) The ablation study discussion can be revisited to add more nuance and emphasize the magnitude of the effects. For instance, not all ConvFormer components are equally important (some cause only a 1% drop in accuracy), and this can be highlighted. The comparison to the patch tokenizer can be moved to the main text from the appendix.
>
> Thank you for the suggestion. Although we noted in L279 that components such as token count, positional/type encoding, and dropout are relatively important, we agree that this could be made clearer. We will revise the ablation study discussion to emphasize the magnitude of each component’s impact and move the comparison of spectra tokenizers to the main text if space allows.
>
> > Not a weakness, but a suggestion: authors can list their key contributions at the end of the intro, as this is typical in NeurIPS papers and helps the reader to follow the paper.
>
> We thank the reviewer for the suggestion, and we will add a key contributions section at the end of the introduction in the camera ready version if space allows.
>
> > How long does it take to train one NMRDiff3D model?
>
> For NMRDiff3D-M, it takes 12h to train on the SpectraBase dataset, 36h on the USPTO dataset, and 26h on the SpectraNP dataset with 4 A100 GPUs. For NMRDiff3D-L, it takes 20h to train on the SpectraBase dataset, 55h on the USPTO dataset, and 58h on the SpectraNP dataset with 4 A100 GPUs. A table of training and sampling time of our NMRDiff3D models on all datasets is found in the Appendix Table 5. In short, the training time increases with the model size, the dataset size, and the size of molecules.
>
> Thank you again for your helpful feedback – we believe the revisions will directly address your concerns and improve the clarity and impact of the paper.
>
> References:
>
> [1] Alberts, Marvin, et al. "Unraveling molecular structure: A multimodal spectroscopic dataset for chemistry." Advances in Neural Information Processing Systems 37 (2024): 125780-125808.
>
> [2] Polykovskiy, Daniil, et al. "Molecular sets (MOSES): a benchmarking platform for molecular generation models." Frontiers in pharmacology 11 (2020): 565644.

---

> > ### Comment · Reviewer_BFWB · 2025-08-08
> > **All concerns resolved**
> >
> > My concerns about the applicability domain are addressed by the authors in the rebuttal by experiments.
> >
> > I thank the authors for the elaborate experiments and explanations.
> >
> > I increased my rating accordingly.

---

> > > ### Author Response · Authors · 2025-08-09
> > >
> > > We thank the reviewer again for the detailed feedback and for recognizing our work!

---

### Official Review · Reviewer_1WdQ · 2025-06-30

**Clarity:** 2
**Significance:** 2
**Originality:** 2
**Rating:** 4
**Confidence:** 3

**Summary:**

The paper introduces NMRDiff3D, a hybrid convolutional transformer and 3D atomic coordinate diffusion model for predicting small molecule structures directly from 1D NMR spectra and chemical formula, achieving state-of-the-art results on both synthetic and semi-real benchmarks.
The authors also present SpectraNP, a large-scale synthetic NMR dataset tailored for natural products, which fills a critical gap in available training data, especially on more complex (i.e., >101 atoms) molecules.

**Questions:**

- Can you provide scaffold-level and fingerprint-based diversity analyses to substantiate the generalization claim, and have you evaluated performance on a scaffold-based split or filtered out scaffold overlaps during testing?
- Can you provide a systematic failure mode analysis, especially on experimental spectra?

**Ethical Concerns:**

["NO or VERY MINOR ethics concerns only"]

**Final Justification:**

The rebuttal addressed the initial concerns I had, and the authors show strong results in all proposed aspects.

**Limitations:**

Yes

**Quality:**

3

**Strengths And Weaknesses:**

Strengths:

- NMRDiff3D achieves state-of-the-art performance on synthetic and semi-real benchmarks and shows consistent prediction quality across a wide range of molecular sizes.
- The paper includes ablation studies isolating the contributions of individual components and attempts to evaluate results on experimental data, although this evaluation is very limited in scale.
- While the model does not handle stereochemistry, this aligns with the protocol followed by prior work (Hu et al., 2023) and is therefore appropriate given the benchmark comparison.

Weaknesses:

- The paper claims that joint training on USPTO and SpectraNP demonstrates generalization across chemically dissimilar datasets. However, this claim remains insufficiently substantiated due to the absence of scaffold-level or fingerprint-based diversity analysis; the fact that only 306 molecules are shared between USPTO and SpectraNP does not preclude significant scaffold or chemotype overlap that could lead to memorization rather than true generalization. As it stands, there paper lacks: (1) Bemis-Murcko scaffold overlap analysis to assess shared core structures and rule out memorization; (2) dissimilarity-based splits or stratified performance metrics to evaluate generalization to unseen scaffolds; and (3) fingerprint similarity statistics or distribution plots (such as Tanimoto similarity histograms of ECFP4 / MACCS keys) between datasets to quantify molecular diversity. Without these, it is unclear whether performance gains arise from genuinely transferable representations or interpolation within familiar chemotypes, undermining the central generalization claim.
- The lack of a systematic categorization of failure types (such as incorrect functional groups or misconnected substructures), or failure mode analysis based on specific compound classes (such as halogenated molecules, sugars, large rings), prevents understanding the boundaries of the model’s reliability and might inflate generalizability claims (like performing well on average classes but failing systematically on important ones). There are also no metrics quantifying the domain shift from synthetic to experimental spectra. A calibration of prediction confidence (e.g., likelihood of correctness across different noise levels, so an evolution of the general confidence score cited in Future Works in the Conclusion) and measurement of accuracy degradation due to factors like solvent noise, peak shifting, or missing atoms would also be appreciated.
- While the authors do state that stereochemistry was excluded from their input/output space and note that Alberts et al [3]. included it, they fail to clearly articulate that this renders the tasks non-equivalent. Including stereochemistry significantly increases the complexity of the structure elucidation problem, and this difference should be explicitly highlighted to avoid misleading comparisons.

---

> ### Author Rebuttal · Authors · 2025-07-31
>
> Thank you for your careful reading and thoughtful feedback. We appreciate your recognition of our model’s strong performance and agree that dataset similarity analysis, generalization experiments to unseen scaffolds, and systematic failure mode analysis would help clarify the scope of our approach and strengthen our manuscript. Below, we provide clarifications and new results addressing these points.
>
> **[Dataset similarity analysis]** Thanks for your question on the similarity between the USPTO and SpectraNP datasets. We compute the Scaffold similarity (Scaff) and fingerprint-based Tanimoto Similarity to a nearest neighbor (SNN) metrics [1] between the test set and training set, as shown in the table below. The two metrics between the SpectraNP test set and the USPTO training set are significantly lower, indicating that the two datasets are relatively dissimilar.
>
> | Test Set | Train set | Scaff(Test, Train) ↑ | SNN(Test, Train) ↑ |
> |-|-|-|-|
> |SpectraNP | SpectraNP | 0.894 | 0.722 |
> |SpectraNP | USPTO | 0.213 | 0.430 |
>
> **[Generalization to unseen scaffolds]** Thanks for your suggestion. We evaluated all models on test subsets with scaffolds unseen during training. As shown in the table, NMRDiff3D still significantly outperforms baselines across these subsets, demonstrating its generalization ability. Moreover, NMRDiff3D-L trained jointly on SpectraNP and USPTO achieves >5% accuracy gain over training on SpectraNP alone, indicating its ability to learn from scaffold-dissimilar datasets.
>
> |Dataset| Model| Top-1| Top-5| Top-10|
> |-|-|-|-|-|
> ||| Acc% ↑ / Sim ↑ | Acc% ↑ / Sim ↑ | Acc% ↑ / Sim ↑ |
> | SpectraBase | Hu et al. | 39.59 / 0.653  | 55.43 / 0.776 | 59.85 / 0.807 |
> | SpectraBase | NMR-DiGress | 43.10 / 0.607 | 59.70 / 0.759 | 64.83 / 0.793 |
> | SpectraBase | NMRDiff3D-M | 65.06 / 0.746 | 75.22 / 0.846 | 78.01 / 0.868 |
> | SpectraBase | NMRDiff3D-L | **68.80 / 0.789** | **77.59 / 0.867** | **80.35 / 0.886** |
> | USPTO | Hu et al. | 15.86 / 0.503 | 23.95 / 0.626 | 26.73 / 0.657 |
> | USPTO | NMR-DiGress | 12.93 / 0.388 | 25.27 / 0.596 | 31.42 / 0.653 |
> | USPTO | NMRDiff3D-M | 74.52 / 0.834 | 84.25 / 0.916 | 86.85 / 0.932 |
> | USPTO | NMRDiff3D-L | **75.75 / 0.844** | **84.71 / 0.918** | **87.05 / 0.933** |
> | SpectraNP | Hu et al.| 6.29 / 0.450 | 13.56 / 0.594 | 17.35 / 0.633 |
> | SpectraNP | NMR-DiGress | 1.59 / 0.222 | 4.50 / 0.389 | 6.18 / 0.436 |
> | SpectraNP | NMRDiff3D-M | 30.78 / 0.473 | 42.27 / 0.645 | 45.66 / 0.686 |
> | SpectraNP | NMRDiff3D-L | 33.30 / 0.504 | 43.04 / 0.655 | 46.31 / 0.692 |
> | SpectraNP | NMRDiff3D-L (JT) | **38.81 / 0.549** | **50.05 / 0.701** | **53.36 / 0.739** |
>
> **[Failure mode analysis]** We agree that a systematic failure mode analysis of molecular structures can help understand our model’s generalization ability. We analyze the failure rate by the number of atoms, number of rings, largest ring size, and functional groups of our NMRDiff3D-L on the synthetic SpectraNP and Real-SpecTeach datasets, as shown in the tables below.  On both datasets, the model fails more often on molecules with the most atoms or the largest number of rings due to less training data and increasing complexity of spectra and structures for larger molecules. However, the model is not systematically significantly failing in a specific category.
>
> **Table: Failure rate by total number of atoms on SpectraNP**
> |# Atoms| ≤40 | 41-80 | 81-120 | 121-160 | 161-200 | >200 |
> |-|-|-|-|-|-|-|
> | # Molecules | 2265 | 6168 | 1819 | 539 | 140 | 22 |
> | Failure Rate | 0.315 | 0.353 | 0.345 | 0.282 | 0.400 | 0.636 |
>
> **Table: Failure rate by number of rings on SpectraNP**
> | # Rings | 0 | 1-2 | 3-4 | 5-6 | 7-8 | >8 |
> |-|-|-|-|-|-|-|
> | # Molecules | 559 | 3071 | 4437 | 2102 | 560 | 224 |
> | Failure Rate | 0.229 | 0.322 | 0.339 | 0.380 | 0.393 | 0.451 |
>
> **Table: Failure rate by largest ring size on SpectraNP**
> | Largest Ring Size | 3-5 | 6 | 7-8 | >8 |
> |-|-|-|-|-|
> | # Molecules | 483 | 8061 | 778 | 1072 |
> | Failure Rate | 0.406 | 0.329 | 0.477 | 0.368 |
>
> **Table: Failure rate by functional group on SpectraNP**
> | Functional Group | Carbonyls | Amides | Esters | Ethers | Alcohols | Halogen | Sulfur | Nitrogen |
> |-|-|-|-|-|-|-|-|-|
> | # Molecules | 8149 | 1480 | 4389 | 8048 | 6862 | 477 | 139 | 3047 |
> | Failure Rate| 0.349 | 0.273 | 0.358 | 0.346 | 0.338 | 0.262 | 0.245 | 0.322 |
>
> **Table: Failure rate by total number of atoms on Real-SpecTeach**
> | #Total Atoms | ≤10 | 11-15 | 16-20 | 21-25 | 26-30 | >30 |
> |-|-|-|-|-|-|-|
> | Total Molecules | 10 | 60 | 76 | 48 | 26 | 18 |
> | Failure Rate | 0.000 | 0.133 | 0.184 | 0.354 | 0.423 | 0.389 |
>
> **Table: Failure rate by number of rings on Real-SpecTeach**
> | # Ring | 0 | 1 | >1 |
> |-|-|-|-|
> | # Molecules | 157 | 69 | 12 |
> | Failure Rate | 0.191 | 0.304 | 0.500 |
>
> **Table: Failure rate by largest ring size on Real-SpecTeach**
> | Largest Ring Size | 3-5 | 6 |
> |-|-|-|
> | # Molecules | 4 | 77 |
> | Failure Rate |0.250| 0.338 |
>
> **Table: Failure rate by functional group on Real-SpecTeach**
> | Functional Group | Carbonyls | Amides | Esters | Ethers | Alcohols | Halogen | Sulfur | Nitrogen |
> |-|-|-|-|-|-|-|-|-|
> | # Molecules | 125 | 3 | 46 | 58 | 43 | 24 | 0 | 24 |
> | Failure Rate| 0.200 | 0.667 | 0.348 | 0.328 | 0.256 | 0.042 | 0.000 | 0.417 |
>
> We also agree that it’s important to quantify the domain shift between synthetic and real spectra. Following [2], we computed the cosine similarity between synthetic and real spectra for all experimental datasets. The table below shows that the Real-SpecTeach is less noisy than our Real-5mer dataset, and the domain shift for 13C spectra is larger than for 1H spectra.
>
> | Dataset | Cos Sim (1H)  | Cos Sim (13C) |
> | --- | --- | --- |
> | Real-SpecTeach | 0.174 ± 0.195 |  0.047 ± 0.072 |
> | Real-5mer-D2O | 0.079 ± 0.063 |  0.030 ± 0.012 |
> | Real-5mer-DMSO | 0.089 ± 0.039 | 0.022 ± 0.034 |
>
> We also report the average cosine similarity of successful and failed predictions on the Real-SpecTeach. The table below shows that failed predictions have lower similarity between synthetic and real 1H spectra. We note that it’s non-trivial to develop systematic metrics to quantify the spectra domain shift, and it is a great suggestion for our future work.
>
> | Prediction | Count | Cos Sim (1H)  | Cos Sim (13C) |
> | --- | --- | --- | --- |
> | Success | 181 | 0.190 ± 0.202 | 0.047 ± 0.072 |
> | Fail | 58 | 0.126 ± 0.166 | 0.048 ± 0.072 |
>
> **[Include stereochemistry]** We agree that including stereochemistry significantly increases the complexity of the structure elucidation problem, and we will highlight this complexity in our revisions. We initially did not include stereochemical information as 2D NMR spectra, degradation analysis, Marfey's analysis or Mosher ester analysis are usually required to determine absolute configuration at chiral centers. For a fair comparison with Alberts et al. [2], we trained and evaluated NMRDiff3D-M on the USPTO including stereochemistry as [2]. The table below shows that our model still consistently outperforms Alberts et al. with single 1H spectra or 1H and 13C spectra with stereochemistry information.
>
> |Spectra|Model| Top-1| Top-5| Top-10|
> |-|-|-|-|-|
> ||| Acc% ↑ / Acc% (w/o stereo) ↑ | Acc% ↑ / Acc% (w/o stereo) ↑ | Acc% ↑ / Acc% (w/o stereo) ↑ |
> | 1H | Alberts et al. | 64.99 / NA | 81.94 / NA | 84.07 / NA |
> | 1H | NMRDiff3D-M | **76.88 / 84.71** | **89.49 / 92.27** | **92.43 / 93.92** |
> | 1H + 13C | Alberts et al. | 73.38 / NA | 87.94 / NA |  89.98 / NA |
> | 1H + 13C | NMRDiff3D-M | **78.08 / 86.18** | **90.27 / 93.05** | **93.05 / 94.54** |
>
> **[Limited evaluation on experimental data]** We thank the reviewer and Reviewer Jfo2 for suggesting additional experimental evaluations beyond our Real-5mer and Real-Spec-Teach datasets. We have now evaluated the zero-shot performance of our model on 23k experimental 13C spectra from the NMRShiftDB2 dataset [3] suggested by reviewer Jfo2, and our method obtains 49.51% top-1 accuracy and 56.08% top-5 accuracy.
>
>
> Thank you again for your helpful feedback – we believe the revisions will directly address your concerns and improve the clarity and impact of the paper.
>
> References:
>
> [1] Polykovskiy, Daniil, et al. "Molecular sets (MOSES): a benchmarking platform for molecular generation models." Frontiers in pharmacology 11 (2020): 565644.
>
> [2] Alberts, Marvin, et al. "Unraveling molecular structure: A multimodal spectroscopic dataset for chemistry." Advances in Neural Information Processing Systems 37 (2024): 125780-125808.
>
> [3] Kuhn, Stefan, and Nils E. Schlörer. "Facilitating quality control for spectra assignments of small organic molecules: nmrshiftdb2–a free in‐house NMR database with integrated LIMS for academic service laboratories." Magnetic Resonance in Chemistry 53.8 (2015): 582-589.

---

### Official Review · Reviewer_Kxkp · 2025-07-02

**Clarity:** 3
**Significance:** 2
**Originality:** 3
**Rating:** 5
**Confidence:** 2

**Summary:**

The paper introduces NMRDiff3D, a diffusion based framework to predict the 3D structure of organic molecules conditioned on the 1D NMR spectra (both H and C spectra) and the chemical formula. The model was trained on a curated set of synthetic data, which is around 100k in size. The paper introduces two metrics about accuracy and similarity and compares their method to two LLM based methods and one graph based method.

**Questions:**

- Can you clarify the technical contributions to the model in the introduction (i.e. what components needed to be changed for this specific problem)?
- If you are generating 3D structures (and have loss terms on the coordinates) why is there no RMSD metric on the atom positions?
- How generalizable is the method across different atom types and different solvents? Are there limitations here?
- Can you elaborate on why graph methods are limited to 64 atoms?

**Ethical Concerns:**

["NO or VERY MINOR ethics concerns only"]

**Final Justification:**

The paper addresses an interesting problem: 3D structure prediction from NMR spectra. The authors successfully apply a diffusion transformer to new problem space. My main concerns were: weak evaluations/baselines and questions about the generalization of the method. Both of these were addressed in the rebuttal. The authors added a number of evaluations/baselines based on my and other review comments. Also, the authors provided additional evidence the model can generalize to most of the solvents used in NRM.

One issue not fully resolved is the framing of equivariance. There wasn’t any direct evidence provided that learning symmetries through data augmentation is better than using a symmetry aware network for the task in this paper, so this design choice should be left as an open question. This is a minor issue that can be resolved in the camera-ready version.

Overall it is a solid contribution so I lean towards acceptance, with the caveat being I’m not expert in NMR and I don't understand the standard techniques used in the field (I'm taking the authors word on that).

**Limitations:**

- Are there better ways to quantify the performance of the method? Can you can compare to any physics based methods or methods that are currently used by the community? If that isn't possible, what is level of accuracy needed to see this method widely adopted? There are lots of reasons to like this paper but without more context/clarity on the impact, it is hard to recommend accept.

**Paper Formatting Concerns:**

I didn't notice any major formatting issues.

**Quality:**

3

**Strengths And Weaknesses:**

Strengths:

- The paper is well written and is of sufficient quality for NeurIPS
- This is a very interesting/relevant problem that not a lot of people focus
- As far as I can tell (I have not specifically worked on this problem), the paper presents a novel diffusion based approach to the inverse problem of spectra to structure

Weaknesses:

- While the idea is novel the pieces used are not, it would be good to clarify what required new thinking vs what was putting together existing pieces
- It is difficult to assess how good/useful the method actually is, the baselines seem quite weak.
- The paper emphasizes a scalable non-equivariant approach but are training on small datasets < 1M in size, being non-equivariant is fine but in the small data regime it might actually be beneficial to enforce symmetries because those models have been shown to be very data efficient. Interesting to note that coord augmentation had the largest impact out of all ablations shown table 3.

---

> ### Author Rebuttal · Authors · 2025-07-31
>
> Thank you for your thoughtful review and feedback. We appreciate your recognition of the novelty of our approach (“novel diffusion-based approach”), the novelty of our problem (“a very interesting/relevant problem that not a lot of people focus”), and the clarity of our paper (“The paper is well written and is of sufficient quality for NeurIPS”). Below, we address your questions and concerns point by point.
>
> **[Clarifications on technical contributions]** As recognized by reviewers 5F4B and BFWB, our method is “the first approach that uses diffusion to predict 3D coordinates from 1D NMR spectra and chemical formula.” While this model builds on diffusion transformers, the novelty lies in applying them to the challenging and underexplored task of structure elucidation from 1D NMR spectra. To do so, we introduce a hybrid convolutional-transformer encoder for 1H and 13C spectra and adapt a non-equivariant diffusion transformer to condition on spectral and formula embeddings for complex natural products. To the best of our knowledge, we are also the first to include a smooth LDDT loss term in the diffusion objective for small molecule generation to improve the local structural fidelity. We will revise the introduction to clarify and highlight these contributions.
>
> **[Weak baselines]** We emphasize that the baselines may appear weak because the task of structure elucidation from 1D NMR spectra is intrinsically challenging and remains largely underexplored. Part of our contribution is to establish stronger modeling approaches for this new and difficult task.
>
> **[Equivariance]** We agree that equivariant models are valuable in low-data regimes. However, we note that datasets on the order of 100k-1M do not necessarily require equivariance for strong performance in 3D atomic generation. For example, MCF [1] shows that a non-equivariant transformer outperforms equivariant baselines for the molecular conformation generation task on GEOM-QM9 (130k) and GEOM-DRUGS (304k) [2]. Recently, a non-equivariant version of Proteina [3] for protein backbone generation was shown to outperform equivariant baselines on Foldseek-AFDB (588k).
>
> **[RMSD 3D metric]** Thank you for the suggestion. We report the top-k Average Minimum RMSD (AMR) of heavy atoms in the table below. We initially did not compute RMSD as there is no baseline of comparison for other non-3D methods, and RMSD is not a size-independent metric, which limits its interpretability, but we find that the AMR on each dataset is reasonable considering the complexity of the dataset. We will add this metric to the appendix of our manuscript.
>
> | Dataset  | Model | Top-1 AMR ↓  | Top-5 AMR ↓ | Top-10 AMR ↓ |
> |-|-|-|-|-|
> | SpectraBase | NMRDiff3D-M | 1.146 | 0.818 | 0.713 |
> | SpectraBase | NMRDiff3D-L | 1.153 | 0.813 | 0.700 |
> | USPTO | NMRDiff3D-M | 1.518 | 1.090 | 0.950 |
> | USPTO | NMRDiff3D-L | 1.525 | 1.094 | 0.954 |
> | SpectraNP | NMRDiff3D-M | 2.108 | 1.666 | 1.533 |
> | SpectraNP | NMRDiff3D-L | 2.106 | 1.666 | 1.539 |
> | Real-SpecTeach | NMRDiff3D-M | 0.885 ± 0.032  | 0.613 ± 0.025 | 0.529 ± 0.019
> | Real-SpecTeach | NMRDiff3D-L | 0.870 ± 0.017  | 0.577 ± 0.002 | 0.503 ± 0.014 |
> | Real-5mer-D2O | NMRDiff3D-M | 2.814 ± 0.050 | 2.194 ± 0.104 | 2.065 ± 0.056 |
> | Real-5mer-DMSO | NMRDiff3D-M | 2.906 ± 0.164 | 2.240 ± 0.065 | 2.025 ± 0.105 |
>
> **[Generalization across atom types and solvents]** Our model is trained on a diverse set of natural products with a wide range of atom types (C, H, O, N, S, P, F, Cl, Br, I) in the CDCl3 solvent, as shown in Appendix Table 2. To assess generalization across different atom types or local environments, we compute the top-10 accuracy of our NMRDiff3D-L by functional group on the SpectraNP test set. As shown in the table below, our model performs consistently across different functional groups.
>
> Additionally, we note that our model trained on the 5mer dataset with CDCl3 solvent can achieve nearly 100% accuracy on the Real-5mer dataset with D2O and DMSO solvents (Figure 4), suggesting its potential to generalize to unseen solvents. However, we acknowledge that performance will not necessarily hold for solvents with different polarities, and will expand the limitations section to reflect this.
>
> **Table: Top-10 accuracy by functional group on the SpectraNP test set.**
> | Functional Group | Carbonyls | Amides | Esters | Ethers | Alcohols | Halogen | Sulfur | Nitrogen |
> |-|-|-|-|-|-|-|-|-|
> | # Molecules | 8149 | 1480 | 4389 | 8048 | 6862 | 477 | 139 | 3047 |
> | Top-10 Accuracy | 0.651 | 0.727 | 0.642 | 0.654 | 0.662 | 0.738 | 0.755 | 0.678 |
>
> **[Limitations of existing graph methods]** Apologies for the confusion. Graph-based methods are not inherently limited to 64 atoms; however, the existing graph methods [4, 5, 6], based on Markov decision processes, beam search, or Monte Carlo tree search, mentioned in the related work section, address molecular graphs with no more than 64 atoms. This may be due to a lack of large-scale spectra datasets for larger molecules or the computational complexity of search-based methods when applied to larger molecules. We will revise the manuscript to clarify this point.
>
> **[Evaluation and practical adoption]** Thank you for your comments about evaluation. There are currently no widely adopted methods that can reliably predict structures from 1D NMR spectra, which was the motivation for this work. The main approach used in the community is manual structure elucidation by expert chemists, which is time-consuming and labor-intensive. We believe our NMRDiff3D model can be used to suggest candidate structures that chemists can then verify, significantly reducing the time and effort involved. While it's hard to define a universal accuracy threshold for full adoption, we believe our current performance is already useful in real application settings.
>
> Thank you again for your careful reading. We believe these clarifications and planned revisions will substantially strengthen the paper.
>
> References:
>
> [1] Wang, Yuyang, et al. "Swallowing the bitter pill: Simplified scalable conformer generation." arXiv preprint arXiv:2311.17932 (2023).
>
> [2] Axelrod, Simon, and Rafael Gomez-Bombarelli. "GEOM, energy-annotated molecular conformations for property prediction and molecular generation." Scientific Data 9.1 (2022): 185.
>
> [3] Geffner, Tomas, et al. "Proteina: Scaling flow-based protein structure generative models." arXiv preprint arXiv:2503.00710 (2025).
>
> [4] Jonas, Eric. "Deep imitation learning for molecular inverse problems." Advances in neural information processing systems 32 (2019).
>
> [5] Huang, Zhaorui, et al. "A framework for automated structure elucidation from routine NMR spectra." Chemical Science 12.46 (2021): 15329-15338.
>
> [6] Sridharan, Bhuvanesh, et al. "Deep reinforcement learning for molecular inverse problem of nuclear magnetic resonance spectra to molecular structure." The Journal of Physical Chemistry Letters 13.22 (2022): 4924-4933.

---

> > ### Comment · Reviewer_Kxkp · 2025-08-05
> >
> > Thank you for your response. A couple follow up comments/questions:
> >
> > **[Equivariance]** Did you try or run ablations with an equivariant architecture?
> >
> > **[RMSD 3D metric]** I assume these are in angstroms?
> >
> > **[Generalization across atom types and solvents]** You evaluated the model on three solvents (CDCl3, D2O, DMSO), how many different solvents are typically used for NMR?

---

> > > ### Author Response · Authors · 2025-08-05
> > >
> > > Thank you for your careful reading and follow-up questions.
> > >
> > > **[Equivariance]** During the early stage of our architecture design, we had initial experiments with EGNN [1], a common equivariant network for small molecule generation, but the results were not promising. For example, in our initial attempt, the percentage of valid generated molecules using EGNN on a 100k dataset with fewer than 19 atoms was ~40%, while the validity using a non-equivariant diffusion transformer was ~67%. However, we acknowledge that investigating scalable equivariant architecture is an interesting future direction in the 3D atomic structure generation field.
> > >
> > > **[RMSD 3D metric]** Thank you for the clarification. Yes, the unit of RMSD is angstroms(Å), and we will clarify this in our revisions.
> > >
> > > **[Generalization across atom types and solvents]** There are typically 7 solvents used for NMR, including CDCl3, D2O, DMSO, (CD3)2CO, CD3CN, CD3OD, and C6D6 [2]. We now report top-10 zero-shot accuracy on ~2k experimental 13C spectra paired with solvent information from the NMRShiftDB2 dataset [3], as part of a further analysis of an experiment suggested by Reviewer Jfo2. As shown in the table below, our model trained on synthetic 13C spectra with CDCl3 solvent shows generalization ability to various solvents.
> > >
> > > | Solvent | # Molecules | Top-10 Zero-shot Accuracy |
> > > |---------|-------------|--------------|
> > > | CDCl3   | 1553        | 0.402        |
> > > | DMSO    | 280         | 0.671        |
> > > | CD3OD   | 216         | 0.338        |
> > > | C6D6    | 49          | 0.204        |
> > > | D2O     | 45          | 0.756        |
> > > | CD3CN   | 17          | 0.353        |
> > > |(CD3)2CO  |  2 |0.500|
> > >
> > > References:
> > >
> > > [1] Satorras, Vıctor Garcia, Emiel Hoogeboom, and Max Welling. "E (n) equivariant graph neural networks." International conference on machine learning. PMLR, 2021.
> > >
> > > [2] Kotlyar, Vadim. "NMR chemical shifts of common laboratory solvents as trace impurities." The Journal of organic chemistry (1997).
> > >
> > > [3] Kuhn, Stefan, and Nils E. Schlörer. "Facilitating quality control for spectra assignments of small organic molecules: nmrshiftdb2–a free in‐house NMR database with integrated LIMS for academic service laboratories." Magnetic Resonance in Chemistry 53.8 (2015): 582-589.

---

> > > > ### Comment · Reviewer_Kxkp · 2025-08-08
> > > >
> > > > **[Equivariance]**
> > > >
> > > > > For example, in our initial attempt, the percentage of valid generated molecules using EGNN on a 100k dataset with fewer than 19 atoms was ~40%, while the validity using a non-equivariant diffusion transformer was ~67%
> > > >
> > > > It is possible that this difference is due to the expressivity of the network rather than the presence or absence of equivariance. EGNN is not known to be the most expressive equivariant GNN. I recommend including some nuance in the text about equivariance for two reasons. First, there were no definitive ablation studies conducted. Second, for the ablations that were performed, coordinate augmentations were by far the most important —so clearly symmetries matter.
> > > >
> > > > **[Generalization across atom types and solvents]**
> > > >
> > > > Thanks for providing more info here, this addresses my concerns.
> > > >
> > > > Besides the minor note on equivariance, all my concerns have been addressed. I will update my score accordingly.

---

> > > > > ### Author Response · Authors · 2025-08-09
> > > > >
> > > > > Thank you for your suggestion. We agree that it is important to use coordinate augmentation to help our diffusion transformer learn equivariance, and we will incorporate a more nuanced discussion about equivariance in our revisions.
> > > > >
> > > > > Thank you again for your detailed feedback and for recognizing our work!

---

### Official Review · Reviewer_5F4B · 2025-07-02

**Clarity:** 4
**Significance:** 3
**Originality:** 3
**Rating:** 5
**Confidence:** 4

**Summary:**

This paper presents a novel, state of the art approach to generating the complete structure of a small molecule using a diffusion model from 1D NMR (Carbon and Hydrogen) experiments and the molecular formula. They outperform a previous model as well as a version of their model that generates graphs. The model is quite promising for aiding biochemists in structure determination from these measurements. They also present a new training dataset, SpectraNP, of predicted 1D NMR measurements generated by MestreNova. So the model is trained with simulated spectra, but they then test it on experimental data with excellent results.

I have read the authors' response, and I am satisfied to keep my score as is.

**Questions:**

Please answer any questions that arose in my review, thanks!

**Ethical Concerns:**

["NO or VERY MINOR ethics concerns only"]

**Final Justification:**

I have read the authors' response, and I am happy to keep my (accept) score. They have addressed all of my concerns.

**Limitations:**

The only obvious limitation is the training set; a larger dataset would lead to better results.

Also, I wonder if MestreNova would object to the public release of a dataset of over 100k molecules' predicted 1D NMR (H & C) that was computed with their commercial software?

**Quality:**

4

**Strengths And Weaknesses:**

Strengths

Quality

This is a high-quality paper. The model is fairly clearly described, the results are very good, they even provide an alternative baseline model that generates graphs rather than 3D coordinates. Their model outperforms Hu et al. and Alberts et al. often by very large amounts (e.g., 86.6 vs. 73.4 Top-1 accuracy on UCPTO).

Clarity

The approach is well-described. They will release their code when the paper is published.

Significance

This is a tool that would gain significant usage by the Natural Products community, if it was posted on the web with a user-friendly interface. Is there a plan to do that?

Originality

There exist models that are end-to-end for 1D NMR that predict structure (their reference 20), but doesn’t use diffusion.
There also exist models that use diffusion to generate 3D structure (their references 19, 44, 65, 38)
but don’t generate molecules conditional on some data, like here, 1D NMR + chemical formula.
So, AFAIK, this is the first model to use 1D NMR plus molecular formula to conditionally generate molecular structure.

Weaknesses:

The paper claims to have created the first and largest 1D NMR database, with over 100k computed spectra:
"To scale to the complex chemical groups found in natural products, we curate SpectraNP, a large-scale dataset of synthetic 1D NMR spectra for 111,193 complex natural products (up to 274 atoms), significantly expanding the chemical complexity of prior datasets (≤101 atoms) [20, 4]."

There are at least two datasets that belie this claim:

1. Your reference [4] has 790k molecules, but (with respect to your claim) only molecules with fewer than 35 heavy atoms were included. You mention 274 atoms, but how many heavy atoms were included?

2. Your reference [60], NP-MRD has 200k 1D spectra, and is available to the public. As one example, NP-MRD provides the predicted 1D NMR spectra of QS-III (C104H168O55), which has 327 atoms. So this seems to have more molecules $and$ a larger number of atoms.

Is there any advantage of SpectraNP over either of these?

It would be relatively easy to get even more NP structures and simulate them from https://coconut.naturalproducts.net/ which has nearly 700k 2D structures of natural products, so this would be a good source for expanding your dataset.

The assumption of molecular formula from mass spec: This is an NP-Hard problem, and the larger the molecule, the more intractable it gets to compute the formula. Anything over 200 heavy atoms (neglecting hydrogen) gets difficult, although the vast majority of natural products are below that (in coconut, at least).

Presumably, MestreNova has some limitations on the use of its software. Is it within the user agreement to make public a dataset of over 100k computed spectra (for each of 1H and 13C) using their software?


Minor comments/typos/wordsmithing:

Equations 1-3: I’m unclear how this enters into the modeling. Why aren’t you just entering the peak coordinates directly, since they are in the data?

Section 3.2: Pardon my ignorance; I don’t work with diffusion models (yet!). Presumably the output is a certain (fixed) dimension, right? (Correct me if I’m wrong). Do you simply have a maximum dimension and just 0-pad it?

Line 134: Are these thresholds annealed during training?

Lines 143-144: You should include these (scaling functions) in the appendix so the paper is self-contained.

Fig 4: Is NMRDiff3D-L covered by the curve for NMRDiff3D-M in the middle and right hand graph?

---

> ### Author Rebuttal · Authors · 2025-07-31
>
> Thank you for your careful reading and extensive feedback. We are especially grateful for your recognition of the technical novelty (“first diffusion-based model to use 1D NMR to generate 3D structure”), performance (“outperforms baselines often by very large amounts”), and potential impact ("would gain significant usage by the Natural Products community”). Below, we provide additional context on the dataset and modeling assumptions, and address your questions point by point.
>
> **[Clarifications on our SpectraNP dataset]** Thank you for raising the important point on the complexity and scope of our SpectraNP dataset. We will revise our wording to clarify that SpectraNP is a large-scale, synthetic 1D NMR dataset specifically curated for machine learning on complex natural products. While the USPTO [1] dataset contains ~790k molecules, these molecules are limited to up to 35 heavy atoms, whereas ours contains molecules up to 130 heavy atoms. Detailed dataset statistics, including the number of data points, heavy atoms, total atoms, atom types, and solvent types, have been provided in Appendix Table 2. Although NP-MRD [2] includes ~200k predicted spectra and covers larger molecules, its synthetic spectra were generated by learning-based tools, not by MestreNova. For example, synthetic 1H spectra in NP-MRD are predicted by PROSPRE [3], an ML model that was trained on only 577 molecules and tested on ~60 molecules, lacking validation at a large scale. In contrast, we use MestreNova to generate synthetic spectra, which is widely considered a gold standard in the chemistry community and uses a combination of machine learning models with rule-based methods.
>
> We appreciate the reviewer’s suggestion to expand SpectraNP using the COCONUT database [4]. We are actively pursuing this direction and have begun simulating additional spectra from COCONUT to further scale the dataset as part of ongoing work.
>
> **[MestreNova license]** Thank you for raising this point. The MestreNova user agreement permits academic use of the software, and we note that the synthetic NMR spectra in the USPTO dataset [1] were also generated using MestreNova. We will investigate the release and license needed in compliance with any usage limitations.
>
> **[Known molecular formula from mass spectroscopy]** Thank you for bringing up this point. We agree that molecular formula determination can be challenging for very large molecules and will make a note of this in the limitations. We note that, in practice, high-resolution mass spectrometry is commonly combined with isotopic abundance and distribution patterns to infer molecular formulas with high confidence. We will note this in our discussion.
>
>
> **[Minor questions]**
>
> > Equations 1-3: I’m unclear how this enters into the modeling. Why aren’t you just entering the peak coordinates directly, since they are in the data?
>
> The input to our model is the raw 10,000-dimensional NMR spectra vector, where each dimension corresponds to a chemical shift bin and its value represents peak intensity. We use the raw spectra rather than annotated peak coordinates because intensities carry important structural information. For example, in 1H NMR, the peak area reflects proton count. Additionally, using the raw spectra enables end-to-end learning without relying on peak-picking algorithms or external annotation. Further details are provided in Appendix Table 1.
> > Section 3.2: Can you clarify the output dimension of your diffusion transformer?
>
> The diffusion transformer outputs predicted atom coordinates $\hat{\mathbf{X}}_0 \in \mathbb{R}^{N \times 3}$, where $N$ is the maximum number of atoms in the dataset, and 3 represents the 3D coordinates. For example, in SpectraNP, $N=274$, and molecules with fewer atoms are padded with zeros.
>
> > Line 134: Are these thresholds in the smooth LDDT loss annealed during training?
>
> No, the thresholds in the smooth LDDT loss remain fixed during training.
>
> > Lines 143-144: You should include the scaling functions in the appendix so the paper is self-contained.
>
> We have included the scaling functions in Appendix L29, and we are happy to clarify this in L143 of the main paper in our revisions.
>
> > Fig 4: Is NMRDiff3D-L covered by the curve for NMRDiff3D-M in the middle and right-hand graphs?
>
> No. The middle and right-hand graphs in Figure 4 show the zero-shot performance of NMRDiff3D-M on Real-5mer-D2O and Real-5mer-DMSO datasets. Since NMRDiff3D-M achieves near 100% accuracy on these datasets, we did not train NMRDiff3D-L for computational efficiency. We will clarify this in Figure 4 in our revisions.
>
> > This is a tool that would gain significant usage by the Natural Products community if it were posted on the web with a user-friendly interface. Is there a plan to do that?
>
> Yes, we plan to release a user-friendly web interface for NMRDiff3D and the SpectraNP dataset in future work.
>
> Thank you again for your helpful feedback. We believe the revisions will address your concerns and improve the clarity and impact of the paper.
>
> References:
>
> [1] Alberts, Marvin, et al. "Unraveling molecular structure: A multimodal spectroscopic dataset for chemistry." Advances in Neural Information Processing Systems 37 (2024): 125780-125808.
>
> [2] Wishart, David S., et al. "The natural products magnetic resonance database (NP-mrd) for 2025." Nucleic Acids Research 53.D1 (2025): D700-D708.
>
> [3] Sajed, Tanvir, et al. "Accurate prediction of 1H NMR chemical shifts of small molecules using machine learning." Metabolites 14.5 (2024): 290.
>
> [4] Venkata Chandrasekhar, Kohulan Rajan, Sri Ram Sagar Kanakam, Nisha Sharma, Viktor Weißenborn, Jonas Schaub, Christoph Steinbeck, COCONUT 2.0: a comprehensive overhaul and curation of the collection of open natural products database, Nucleic Acids Research, 2024; gkae1063

---

> > ### Comment · Reviewer_Jfo2 · 2025-08-08
> >
> > One comment on the data licensing: In the past, they did not impose any licensing restrictions when releasing data generated with the software. However, this is not clear from their licensing terms and we had to contact them to be able to publish data from MestreNova. It would be good to reach out to them and clarify if this is still their position.

---

> > > ### Author Response · Authors · 2025-08-09
> > >
> > > Thank you again for bringing the data licensing concern to our attention. We have contacted the MestreNova team to clarify their current policy and will investigate it carefully once we receive their reply. We note that the synthetic spectra datasets SpectraBase [1] and USPTO [2] released recently were generated with MestreNova and remain publicly available online now, suggesting that such data sharing for academic use may be permitted in practice.
> > >
> > > References:
> > >
> > > [1] Hu, Frank, et al. "Accurate and efficient structure elucidation from routine one-dimensional NMR spectra using multitask machine learning." ACS Central Science 10.11 (2024): 2162-2170.
> > >
> > > [2] Alberts, Marvin, et al. "Unraveling molecular structure: A multimodal spectroscopic dataset for chemistry." Advances in Neural Information Processing Systems 37 (2024): 125780-125808.

---

### Official Review · Reviewer_Jfo2 · 2025-07-03

**Clarity:** 2
**Significance:** 3
**Originality:** 3
**Rating:** 5
**Confidence:** 4

**Summary:**

In this paper, the authors tackle the challenge of predicting the chemical structure from the combined 1H and 13C-NMR spectra. To this end the authors develop a hybrid diffusion framework consisting of a Transformer based on encoder (with the spectra ingested via patches) and a diffusion transformer generating the molecular structure. Both 1H and 13C spectra are passed into the model as 10,000 dimensional vectors and the molecule is generated as a 3D structure. The authors train and evaluate their framework on synthetic data as well as perform evaluation on experimental spectra. Across all experiments the authors model outperform baselines.

**Questions:**

- The authors modelling approach should work equally well when only applied to 1H or 13C NMR spectra. Is there a reason why only the combination of 1H and 13C was considered? Training the model on only one type of spectra would make benchmarking easier while making the paper more convincing.

- The paper would highly benefit from further benchmarking on experimental data as well as an expanded discussion of the performance on experimental data. Both [1] and [2] evaluate their modelling approaches on experimental 13C data from the nmr db2 [3]. While this database only provides peak positions the data could be easily converted into vector form. [4] also shows performances on the van Bramer dataset and may prove an interesting model to benchmark against.

I am willing to increase my score if my concerns above are sufficiently addressed.

[1]: J. Phys. Chem. Lett. 2022, 13, 22, 4924–4933
[2]: Jonas, NeurIPS 2019
[3]: https://nmrshiftdb.nmr.uni-koeln.de/#:~:text=nmrshiftdb2%20is%20a%20NMR%20database,spectra%2C%20structures%20and%20other%20properties.
[4]: Alberts 2025, DOI: 10.26434/chemrxiv-2025-q80r9

**Ethical Concerns:**

["NO or VERY MINOR ethics concerns only"]

**Final Justification:**

The authors have addressed all my concerns with additional experiments. Accordingly I have raised my score to 5.

**Limitations:**

yes

**Paper Formatting Concerns:**

References out of order.

**Quality:**

2

**Strengths And Weaknesses:**

Strengths:
- Very pertinent topic with high impact on chemistry
- Novel modelling approach
- Extensive ablations

Weaknesses:
- Limited experimental evaluations and benchmarking against existing models

---

> ### Author Rebuttal · Authors · 2025-07-31
>
> We thank the reviewer for their thoughtful review and feedback. We appreciate the reviewer for recognizing the high impact of our work in chemistry, novel modeling approach (similar to reviewers 5f4b and bfwb), and extensive ablations, with reviewer bfwb agreeing that "experimental pipelines are well thought out and executed." Below, we address your questions and concerns point by point.
>
> **[Train on only 1H or 13C spectra]** We agree with your point that our model should, in principle, work well when only applied to 1H or 13C NMR spectra. We have analyzed the impact of training on 1H and/or 13C spectra in Appendix E.3 and Appendix Figure 2. We found that while the model has performed well with a single spectrum type, the combination of 1H and 13C spectra provides complementary information that further improves performance in general. We chose to focus on the combined setting to demonstrate the full potential of our approach.
>
> To benchmark our model against baselines on a single type of spectrum, we evaluate all models on the SpectraBase dataset as shown in Table 1. Our model still consistently and significantly outperforms the baselines with only $^1$H or ${}^{13}$C spectra.
>
> **Table 1: Performance on SpectraBase dataset using $^1$H or ${}^{13}$C spectra.**
>
> | Spectrum Type      | Model         | Top-1  | Top-5 | Top-10  |
> |----------------|--------------|---------|---------|----------|
> |                |              | Acc% ↑ / Sim ↑ | Acc% ↑ / Sim ↑ | Acc% ↑ / Sim ↑ |
> | ${}^{13}$C   | Hu et al.  | 4.50 / 0.296 | 12.92 / 0.460 | 18.27 / 0.523  |
> | ${}^{13}$C  | NMR-DiGress  | 10.87 / 0.314 | 19.35 / 0.438 | 23.00 / 0.477  |
> | ${}^{13}$C  | NMRDiff3D-M | 53.11 / 0.655 | 64.46 / 0.768 | 68.13 / 0.797  |
> | ${}^{13}$C  | NMRDiff3D-L | **57.07 / 0.693** | **67.43 / 0.791** | **70.85 / 0.817**  |
> | $^1$H        | Hu et al. | 31.12 / 0.569 | 48.64 / 0.725 | 54.92 / 0.771  |
> | $^1$H        | NMR-DiGress | 31.13 / 0.521 | 48.73 / 0.682 | 54.52 / 0.723  |
> | $^1$H        | NMRDiff3D-M | 67.29 / 0.768 | 77.87 / 0.862 | 80.91 / 0.884  |
> | $^1$H        | NMRDiff3D-L | **67.87 / 0.779** | **78.39 / 0.868** | **81.59 / 0.890**  |
>
> **[Additional experimental benchmarks]** Thank you for the suggestion for benchmarking on the NMRShiftDB2 dataset [3] used by [1] and [2]. We have evaluated the zero-shot performance of our model on the 23k experimental 13C spectra from NMRShiftDB2, and our method obtains **49.51% top-1 accuracy and 56.08% top-5 accuracy**. In comparison, [1] reports <1% top-5 accuracy on a subset of 2k molecules, and [2] reports 55.9% top-1 accuracy and 58.6% top-5 accuracy on a subset of simpler molecules with only C, H, O, and N elements and fewer than 64 atoms. We further note that [2] requires additional information on the number of adjacent hydrogens bonded to each carbon.
>
> Thank you for pointing out reference [4], which has also been evaluated on the van Bramer dataset (referred to as “Real-SpecTeach” in our manuscript). [4] is a sequence-based approach that reports 53.16% top-1 accuracy with 13C spectra, and 31.58% top-1 accuracy with 1H spectra. In comparison, our NMRDiff3D-L achieves **66.86% top-1 accuracy with 13C spectra and 62.26% top-1 accuracy with 1H spectra**. We note that they evaluate on a subset of the dataset (171 molecules) while we use all 239. Finally, [4] was released on ChemRxiv after the NeurIPS 2025 submission deadline, and their model is not yet fully open-sourced, however we will include a head-to-head benchmark against this method in our revisions.
>
> Thank you again for your helpful feedback. We believe the revisions will directly address your concerns and improve the clarity and impact of the paper.
>
> References:
>
> [1] Sridharan, Bhuvanesh, et al. "Deep reinforcement learning for molecular inverse problem of nuclear magnetic resonance spectra to molecular structure." The Journal of Physical Chemistry Letters 13.22 (2022): 4924-4933.
>
> [2] Jonas, Eric. "Deep imitation learning for molecular inverse problems." Advances in neural information processing systems 32 (2019).
>
> [3] Kuhn, Stefan, and Nils E. Schlörer. "Facilitating quality control for spectra assignments of small organic molecules: nmrshiftdb2–a free in‐house NMR database with integrated LIMS for academic service laboratories." Magnetic Resonance in Chemistry 53.8 (2015): 582-589.
>
> [4] Alberts, Marvin, Nina Hartrampf, and Teodoro Laino. "Automated Structure Elucidation at Human-Level Accuracy via a Multimodal Multitask Language Model." (2025).

---

> > ### Comment · Reviewer_Jfo2 · 2025-08-04
> >
> > Thank you the extensive additional evaluations. All my concerns and questions were addressed and I will raise my score accordingly.

---

> > > ### Author Response · Authors · 2025-08-05
> > >
> > > Thank you again for your constructive feedback and recognition of our work!

---

### Note · Authors · 2025-08-15

We thank the AC and reviewers for engaging with our work. We appreciate that the reviewers recognize the novelty and high impact of the problem in chemistry (Jfo2, 5F4B, Kxkp), our novel modeling approach (5F4B, BFWB, Jfo2, Kxkp), our strong performance (5F4B, BFWB, 1WdQ), and the clarity of the paper (5F4B, Kxkp, BFWB). Our extended experiments and clarifications during the rebuttal period further demonstrate our method’s generalization ability to elucidate unseen 3D structures from large-scale synthetic and experimental NMR spectra, addressing all major reviewer concerns. We will incorporate all reviewers’ helpful feedback into our revised manuscript.

---

### Decision · Program_Chairs · 2025-09-17

**Decision:**

Accept (poster)

**Comment:**

The paper introduces  NMRDiff3D, a diffusion-based framework that directly predicts 3D molecular structures from 1D NMR spectra and molecular formulas. The work is technically novel, addresses an underexplored problem in chemistry, and demonstrates consistent state-of-the-art performance across synthetic and experimental datasets, supported by extensive ablations and additional evaluations after rebuttal. While questions remain around equivariance and dataset scope, the authors provided convincing clarifications and new experiments, leaving the community with a practical tool that can accelerate small-molecule discovery.